# THE LATENT ROAD TO ATOMS: BACKMAPPING COARSE-GRAINED PROTEIN STRUCTURES WITH LATENT DIFFUSION

## ABSTRACT

Coarse-grained(CG) molecular dynamics simulations offer computational efficiency for exploring protein conformational ensembles and thermodynamic properties. Though coarse representations enable large-scale simulations across extended temporal and spatial ranges, the sacrifice of atomic-level details limits their utility in tasks such as ligand docking and protein-protein interaction prediction. *Backmapping*, the process of reconstructing all-atom structures from coarse-grained representations, is crucial for recovering these fine details. While recent machine learning methods have made strides in protein structure generation, challenges persist in reconstructing diverse atomistic conformations that maintain geometric accuracy and chemical validity. In this paper, we present Latent Diffusion Backmapping (LDB), a novel approach leveraging denoising diffusion within latent space to address these challenges. By combining discrete latent encoding with diffusion, LDB bypasses the need for equivariant and internal coordinate manipulation, significantly simplifying the training and sampling processes as well as facilitating better and wider exploration in configuration space. We evaluate LDB's state-of-the-art performance on three distinct protein datasets, demonstrating its ability to efficiently reconstruct structures with high structural accuracy and chemical validity. Moreover, LDB shows exceptional versatility in capturing diverse protein ensembles, highlighting its capability to explore intricate conformational spaces. Our results position LDB as a powerful and scalable approach for backmapping, effectively bridging the gap between CG simulations and atomic-level analyses in computational biology.

## 1 INTRODUCTION

Coarse-Grained Molecular Dynamics (CG-MD) simulation has become an indispensable tool in computational biology for simulating large biomolecular systems (Das & Baker, 2008; Liwo et al., 2014; Kmiecik et al., 2016; Souza et al., 2021; Majewski et al., 2023; Arts et al., 2023). Through grouping atoms into super-atoms or beads, CG models significantly decrease computational requirements and allow the observation of long-time processes such as folding, aggregation, and self-assembly (Lequieu et al., 2019; Shmilovich et al., 2020; Mohr et al., 2022). However, CG representations inherently sacrifice atomistic details of protein structures, limiting their application to a bunch of important downstream tasks in drug discovery, such as molecular recognition, signaling pathways deciphering, and allosteric sites prediction (Badaczewska-Dawid et al., 2020; Vickery & Stansfeld, 2021; Zambaldi et al., 2024). Under such circumstances, *backmapping*, i.e., reconstructing all-atom structures from CG representations, is essential for a comprehensive understanding and wider applications of CG-MD (Huang et al., 2016; Śledź & Caflisch, 2018; Peng et al., 2019; Kim, 2023).

Two primary challenges are faced with backmapping coarse-grained protein representations to all-atom structures. The first challenge is the high dimensionality involved in modeling large biomolecules. Proteins, in particular, consist of thousands of atoms and intricate structural patterns, making it difficult for models to learn and extract relevant features effectively (Rogers et al., 2023; Fu et al., 2024; Wuyun et al., 2024). This complexity also leads to issues during sampling, where directly generating 3D coordinates for numerous atoms can result in chemically invalid structures,

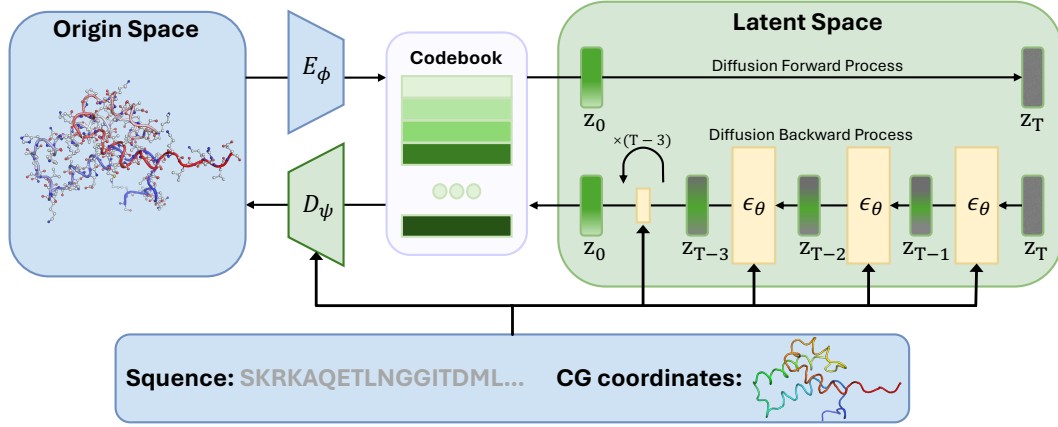

Figure 1: The overall framework of our Latent Diffusion Backmapping (LDB) method. After VQ-VAE training, the protein structure is encoded into a discrete low-dimensional representation without graph structure. The latent vector $\mathbf{z}$, after being perturbed with noise, is passed to the denoising network $\epsilon_\theta$, conditioned on a CG graph structure. The noisy sample $\mathbf{z}_t$ is progressively denoised, and in the final decoding step, the CG structure guides the reconstruction of the full-atom representation.

such as bond length violations and incorrect valency, thus compromising the physical and chemical fidelity of the protein models (Luo & Ji, 2022; Qiang et al., 2023).

The second challenge is the vast and dynamic conformational space that proteins can occupy. Such dynamics result in unique conformational changes, which play a critical role in enabling the diverse functions of proteins and are essential for maintaining the proper physiological functions of living organisms (Miller & Phillips, 2021). Though CG simulations allow us to observe and study these special conformations across temporal and spatial scales, they make an obstacle for structure backmapping. When generative models are provided with multiple CG representations that are topologically similar, the models must not only distinguishing among these simplified inputs, but also reconstruct the all-atom variations in the 3D conformational space with structural and chemical accuracy (Yang & Gómez-Bombarelli, 2023).

Traditional backmapping methods often rely on heuristics to generate initial structures, but these approaches frequently result in non-physical artifacts and fail to capture the thermodynamic diversity of protein conformations (Nicholson & Greene, 2020). Early machine learning approaches, such as generative adversarial networks (Li et al., 2020; Stieffenhofer et al., 2020; 2021; Shmilovich et al., 2022) and variational autoencoders (Wang & Gómez-Bombarelli, 2019; Wang et al., 2022; Yang & Gómez-Bombarelli, 2023), align all-atom structures with the prior distribution of coarse-grained models. However, such methods typically approximate only the most probable conformations and struggle to capture the complex dynamics of structural distributions (Murphy, 2012; Yang & Gómez-Bombarelli, 2023).

Denoising diffusion models (Ho et al., 2020) offer a stochastic approach for sampling protein ensembles. Diffusion over local structural relationships, like bond angles, often requires complex approximations and post-processing for structural validity (Jing et al., 2022; Yim et al., 2023). Latent space methods, while promising, handle both node and edge features, constraining network design and limiting their use to small molecules or backbone-only structures (Xu et al., 2023; Fu et al., 2024). DiAMONDBack (Jones et al., 2023) uses an autoregressive approach to backmap atom coordinates sequentially, but this complicates sampling, reducing both efficiency and quality, particularly with large biomolecules.

In this paper, we propose Latent Diffusion Backmapping (LDB) to address the above challenges. LDB begins by encoding the all-atom structures into a node-level latent representation, capturing equivariance and local structural relationships. By applying physical constraints such as bond lengths and angles, the method ensures chemical validity, thereby eliminating the need for extensive post-processing. Furthermore, the node-level representation allows for greater flexibility in the denoising network architecture, removing the requirement for explicit edge modeling. Finally, LDB

translates these embeddings into discrete, low-dimensional codes, reducing the dimensionality of the generative task and enabling a more efficient and stable training process.

To further improve modeling precision and diversity, LDB incorporates a conditional diffusion model that operates in the discrete latent codes. By introducing conditional diffusion, we enhance the exploration of the latent space, allowing the model to generate diverse and valid conformations while maintaining high accuracy.

We evaluate LDB on widely-used protein dynamics datasets PED (Ghafouri et al., 2024), demonstrating its state-of-the-art performance in reconstructing conformations with high fidelity and chemical correctness. Further experiments on large protein dynamic datasets ATLAS (Vander Meersche et al., 2024) and static proteins datasets PDB (Berman et al., 2000) highlight LDB's superior ability to model protein ensembles, showcasing its potential for practical applications in computational biology.

Our contributions are as follows:

- We introduce LDB, a novel approach designed to address the challenge of limited exploration in conformational space, enabling accurate reconstruction of all-atom 3D protein structures from coarse-grained representations.

- Our method leverages discrete, low-dimensional latent representations that capture structural relationships with inherent equivariance, simplifying the diffusion process and improving overall efficiency.

- By integrating these latent representations with diffusion, our approach significantly enhances structural accuracy and chemical fidelity, making it a robust solution for protein backmapping across diverse datasets.

## 2 RELATED WORK

**Traditional Methods.** Traditional backmapping methods utilize rule-based heuristics to generate initial atomic structures (Lombardi et al., 2016), which are subsequently refined through geometric optimization or energy minimization (Vickery & Stansfeld, 2021). However, these approaches often result in non-physical imperfections, such as atomic clashes and abnormal bond angles (Xu et al., 2019), and the refinement process can be computationally expensive and biased toward specific minimization schemes (Badaczewska-Dawid et al., 2020). Additionally, these methods are deterministic and do not capture the thermodynamic diversity of atomic structures that correspond to a single CG representation (Yang & Gómez-Bombarelli, 2023).

**Data-driven Methods.** Data-driven approaches aim to overcome these limitations by predicting atomic structures from CG representations. While deterministic models like MLPs (An & Deshmukh, 2020) and SE(3)-Transformers (Heo & Feig, 2023) offer high precision, they struggle with the one-to-many nature of backmapping, leading to reduced structural diversity. Chennakesavalu & Rotskoff (2024) uses Gaussian Mixture Models (GMMs) for local rotamer states and a prediction model for global coupling to generate protein conformations. Compared to direct distribution-learning models, it relies more on physical constraints and statistical models, lacking end-to-end optimization of the target distribution, resulting in lower accuracy.

Generative models, including GANs (Li et al., 2020; Stieffenhofer et al., 2020; 2021; Shmilovich et al., 2022) and VAEs (Wang & Gómez-Bombarelli, 2019; Wang et al., 2022; Yang & Gómez-Bombarelli, 2023), address these challenges by learning multimodal distributions of atomic structures. However, GANs are often ineffective at modeling complex distributions, and VAEs tend to prioritize common structures, limiting their ability to generate diverse conformations.

Recent work has shown that diffusion models, such as those proposed by Li et al. (2024) and Jones et al. (2023), are particularly effective for backmapping. These models condition on CG inputs to generate diverse and detailed atomic structures. However, diffusion in atomic space suffers from high computational cost and limited flexibility, particularly for large systems. Moreover, the excessive freedom in exploration can lead to generated structures that deviate from the target conformations.

## 3 BACKGROUND

### 3.1 PROBLEM DEFINITION

**Notations**: Consider an all-atom protein structure as a set of atoms $AA = \{(x_i, v_i)\}_{i=1}^n$, where $n$ denotes the number of protein atoms. The vector $x = \{x_1, \ldots, x_n\} \in \mathbb{R}^{n \times 3}$ represents their three-dimensional coordinates, and $v$ represents the atomic types of the protein (Guan et al., 2023). The coarse-grained structure of the $AA$ is represented as $CG = \{(X_i, V_i)\}_{i=1}^N$, where $N < n$ and $X = \{X_1, \ldots, X_N\} \in \mathbb{R}^{N \times 3}$ indicates the CG coordinates, with $V \in \mathbb{R}^{N_f}$ denoting the amino acid types. We define the sets $[n]$ and $[N]$ as $\{1, 2, \ldots, n\}$ and $\{1, 2, \ldots, N\}$, respectively. The CG operation is then characterized by a surjective mapping $m : [n] \to [N]$, which assigns each FG atom to a CG atom.

**Internal Coordinates representation**: To reconstruct FG structures from CG models, we utilize an internal coordinate representation that describes the adjacency relationships among points as $\mathcal{T} = \{(d_i, \theta_i, \tau_i)\}_{i=1}^{N \times 13}$, where $d_i$ denotes bond lengths, $\theta_i$ represents bond angles, and $\tau_i$ indicates dihedral angles. For each point, we specifically calculate its relative relationships with neighboring points: the bond length to one neighbor, the angle formed with two neighboring points, and the dihedral angle involving three surrounding points. Residues with fewer than 13 heavy atoms are padded to reach the maximum length of 13 heavy atoms. See Appendix A.7 for further details.

**Problem Definition**: Given a protein's coarse-grained structure, defined by coordinates $X$ and the corresponding amino acid types $V$, the task of protein backmapping is to generate the corresponding all-atom coordinates $x$, where the atom types $v$ are determined by the amino acid sequence. The goal is to learn and efficiently sample from the conditional distribution $p(x \mid X, V)$. In this work, we rely on the $C_\alpha$ atoms as they provide a robust representation of protein-protein interactions and serve as a reliable granularity for reverse mapping, following established methods in the field (Badaczewska-Dawid et al., 2020; Yang & Gómez-Bombarelli, 2023; Jones et al., 2023).

### 3.2 DIFFUSION MODEL FOR CONTINUOUS FEATURES

The Denoising Diffusion Probabilistic Model (DDPM) (Sohl-Dickstein et al., 2015; Ho et al., 2020) is a generative modeling framework that transforms complex data distributions into Gaussian noise through a forward diffusion process and subsequently learns to reverse this process to generate new data samples. This model leverages the principles of diffusion processes and denoising autoencoders to achieve high-quality generative performance.

**Forward Diffusion Process**: Given a data point $x_0 \sim q(x_0)$, the forward diffusion process progressively and independently adds a small amount of Gaussian noise to the data over $T$ time steps. Utilizing the properties of Gaussian distributions, we can express the noise adding process and the distribution of $x_t$ given $x_0$ as:

$$q(x_t \mid x_{t-1}) = \mathcal{N}\left(x_t; \sqrt{1 - \beta_t} x_{t-1}, \beta_t \mathbf{I}\right), q(x_t \mid x_0) = \mathcal{N}\left(x_t; \sqrt{\bar{\alpha}_t} x_0, (1 - \bar{\alpha}_t)\mathbf{I}\right), \quad (1)$$

where $\beta_t \in (0, 1)$ is a predefined variance schedule controlling the noise amount added at each step, $\mathbf{I}$ is the identity matrix, $\alpha_t = 1 - \beta_t$ and $\bar{\alpha}_t = \prod_{s=1}^t \alpha_s$ is the cumulative product up to time $t$. As $t$ approaches $T$, the distribution of $x_t$ converges to a standard normal distribution due to the cumulative effect of the added noise.

**Reverse Diffusion Process**: The reverse diffusion process aims to recover $x_0$ from $x_T$ by sequentially removing the added noise. This process is also modeled as a Markov chain but with learned parameters:

$$p_\theta(x_{t-1} \mid x_t) = \mathcal{N}\left(x_{t-1}; \mu_\theta(x_t, t), \sigma_t^2 \mathbf{I}\right), \quad (2)$$

where $\mu_\theta(x_t, t)$ is a neural network parameterized by $\theta$, predicting the mean of the reverse transition, and $\sigma_t^2$ is the variance, often set to $\beta_t$ or learned separately.

**Training Objective**: To streamline the learning process, up-to-date methods (Ho et al., 2020) usually parameterize $\mu_\theta(x_t, t)$ with the noise component at $t$ timestep with $\epsilon_\theta(x_t, t)$, and train the denoising model $\epsilon_\theta$ by minimizing the variational bound on the negative log-likelihood:

$$\mu_\theta(x_t, t) = \frac{1}{\sqrt{\alpha_t}}\left(x_t - \frac{\beta_t}{\sqrt{1 - \bar{\alpha}_t}} \epsilon_\theta(x_t, t)\right), L(\theta) = \mathbb{E}_{x_0, \epsilon, t}\left[\|\epsilon - \epsilon_\theta(x_t, t)\|^2\right]. \quad (3)$$

## 4 METHOD

In this section, we introduce the proposed Latent Diffusion Backmapping (LDB) framework. Our work is inspired by the success of Stable Diffusion (Rombach et al., 2022), which has demonstrated the effectiveness of generating high-resolution images in latent space. However, extending this concept to complex protein structures presents unique challenges (Winter et al., 2022; Xu et al., 2023; Hayes et al., 2024). We address these challenges by first compress the complex all-atom protein structure into **discrete latent codes**, and then apply conditional diffusion in the latent space.

In the following sections, we detail the design of the discrete latent encoding and latent diffusion in Sections 4.1 and 4.2, respectively. An overview of the framework is provided in Figure 1.

### 4.1 DISCRETE LATENT AUTOENCODING

We designed a node-level latent representation to efficiently compress and represent protein structures, as shown in Figure 2. Unlike traditional methods that extract both node and edge features, our approach focuses solely on node-level representations, improving flexibility and reducing complexity. This allows the diffusion model to avoid simultaneous processing of noise addition and removal for both nodes and edges, simplifying the architecture.

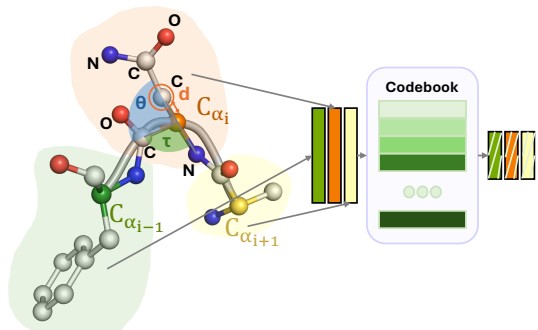

To construct this latent space, we treat each amino acid as a minimal compression unit, reducing the dimensionality of full-atom structures. Given the invariance of protein structures under geometric transformations like rotation and translation, we employ an SE(3)-equivariant graph neural network within the GenzProt (Yang & Gómez-Bombarelli, 2023) framework to extract robust node-level representations.

Figure 2: Illustration of protein structure to discrete latent codes. The all-atom structure of three adjacent residues is encoded into a latent space, capturing their relative spatial relationships. Each residue is mapped to a latent code, which is further compressed and discretized via a codebook, yielding a lower-dimensional representation.

We used internal coordinates as training targets for autoencoder, which include bond lengths and angles, ensuring physical consistency in reconstructed structures. This approach is particularly suited for backmapping tasks, as it reconstructs full-atom structures from coarse-grained representations.

Due to the challenges posed by the limited availability and imbalance in protein conformation data—where some proteins have abundant dynamic structure data while others are represented by only a few or even a single static structure—we chose to employ a Vector Quantized Variational Autoencoder (VQ-VAE) (Van Den Oord et al., 2017). Its ability to discretize continuous features into a fixed-size codebook makes it particularly suited to learning robust representations from such unevenly distributed datasets.

To further enhance efficiency, we compressed the latent representation by mapping it to a lower-dimensional space. This decouples the code lookup from the high-dimensional embedding, allowing for the retrieval of latent variables in a lower-dimensional space, which are then projected back into the original embedding. This method improves the training and diffusion processes.

The encoder $E_\phi$ encodes the all-atom structure into latent space $z$, preserving rotation and translation consistency. The latent variables are quantized via a codebook, and the decoder $D_\psi$ generates internal coordinates, which are used to reconstruct the full-atom structure based on predefined anchor points, following the hierarchical placement algorithm described by (Jing et al., 2022).

### 4.2 GRAPH LATENT DIFFUSION

In this section, we describe the noise addition and removal processes in the latent space, as derived earlier. Unlike traditional diffusion models that operate in high-dimensional coordinate space, our

approach simplifies diffusion by leveraging a lower-dimensional discrete latent codes, avoiding the complexity of geometric parameters, as shown in Figure 1.

Traditional diffusion methods face challenges when applied to protein structures, as they often operate in three-dimensional space or rely on relative distances and angles. This increases complexity and makes it harder to capture the symmetry and physical constraints inherent in protein structures. Moreover, performing diffusion in high-dimensional spaces complicates the multi-step denoising process, making it difficult to accurately model subtle conformational differences.

To overcome these challenges, we focus on node-level latent representations, which embed the necessary structural information. This eliminates the need for explicit geometric constraints, simplifying the noise addition and removal processes. By performing diffusion in the latent space, our method avoids the intricacies of handling node and edge features, resulting in a more streamlined and efficient model.

Additionally, by compressing the latent representation into a discrete code, we mitigate the computational complexity associated with large protein structures. This compact representation allows for efficient diffusion, reducing noise accumulation and improving overall computational efficiency.

We build our denoising network $\epsilon_\theta$ on the ProteinMPNN framework Dauparas et al. (2022), focusing on CG discrete latent codes without modeling edge information, which enhances flexibility. The network processes three inputs: coarse-grained graph node coordinates, residue types, and an initial noise vector. The node coordinates and residue types represent the coarse-grained protein structure and serve as conditional information to refine the noise vector during the denoising process iteratively.

To account for varying noise levels, we modify the LayerNorm layer of ProteinMPNN to adaptive layer norm (adaLN) (Perez et al., 2018), allowing dynamic adjustments during the denoising process. This ensures consistent, physically plausible protein structures across all time steps.

The denoising objective minimizes the difference between predicted and actual noise, as described by the following loss function:

$$L_{\text{diffusion}} = \mathbb{E}_{z_0, \epsilon, t} \left[ |\epsilon - \epsilon_\theta(z_t, t, c)|^2 \right]$$

where $z_0$ is the initial latent variable, $z_t$ is the noisy latent variable at time step $t$, $\epsilon$ represents noise, and $c$ includes conditional information such as graph structure and residue types. This objective enables efficient, accurate denoising while maintaining geometric and chemical consistency.

By embedding symmetry and equivariance in the node-level latent space, our method avoids handling complex physical constraints explicitly, significantly enhancing both the simplicity and computational efficiency of the diffusion process.

## 5 EXPERIMENT

In this section, we evaluate LDB across three diverse protein datasets to demonstrate its broad applicability. (1) On the widely-used PED benchmark (Lazar et al., 2021; Ghafouri et al., 2024), which contains approximately 100 frames with each of the 85 proteins, LDB achieved state-of-the-art (SOTA) structural and chemistry accuracy in reconstructing protein structures. (2) On the larger ATLAS dataset (Vander Meersche et al., 2024), comprising 300 conformations with each of the 1297 proteins, LDB exhibits superior performance in generating diverse protein ensembles, showcasing its capability in capturing conformational variability. (3) Finally, We demonstrate LDB's ability to generalize across the extensive PDB dataset (Berman et al., 2000), containing 62,105 real-world, single-conformation proteins, highlighting its potential for practical backmapping applications. These results collectively underscore the robustness and versatility of the proposed method. For detailed descriptions of the datasets and preprocessing steps, please refer to Appendix A.1.

### 5.1 EXPERIMENTAL SETTINGS

**Baselines.** We selected two recent SOTA backmapping methods as our baselines: GenZProt (Yang & Gómez-Bombarelli, 2023), and DiAMoNDBack (Jones et al., 2023). GenZProt is based on the VAE framework, which employs two encoders to map full-atom and coarse-grained structures into

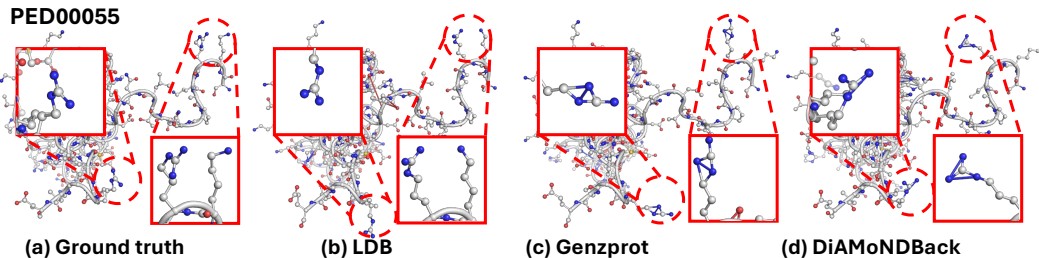

**(a) Ground truth**     **(b) LDB**     **(c) Genzprot**     **(d) DiAMoNDBack**

Figure 3: Visualization of PED00055 protein structure generation from the PED dataset. Our method (b) maintains accurate structural integrity near flexible side chains (red circles), closely matching the ground truth (a). In contrast, Genzprot (c) and DiAMoNDBAck (d) generate conflicting side chain atoms in these regions. See Appendix A.5 for further details.

a latent space, aligning the two representations. Due to its learning mechanism, the model's learned prior distribution does not extend into low-probability regions, limiting its ability to capture the full diversity of protein ensembles. DiAMoNDBack utilizes a diffusion-based framework that defines an auto-regressive structure generation process, leading to the accumulation of errors and significant computational demands. We reproduce all baseline methods following their experimental settings.

**Evaluation Metrics.** We evaluate the generated structures based on two key aspects: (1) structural accuracy, i.e., the similarity of the generated conformations to the original fine-grained structures, where Root Mean Squared Distance (RMSD) and Graph Edit Distance (GED) are applied. (2) chemical validity, i.e., the extent to which the generated structures adhere to realistic chemical properties and constraints, such as bond lengths and angles. Specifically, Steric Clash Score, Interaction Score, and Graph Difference Ratio (GDR) are employed to assess chemical validity. See appendix A.2 for detailed description for these metrics.

**Model Implementations** For the autoencoder in LDB, we adopted the parameter settings from Genzprot, setting the dimensionality of the output node features to 36. The vector quantization employs a codebook size of 4096 with an embedding dimension of 3. Learning rate reduction and early stopping were controlled based on validation loss. The network was trained with a batch size of 4 and an initial learning rate of 0.001 for the PED and PDB datasets, while using 0.0005 for the ATLAS dataset

For the diffusion framework, we used a linear variance schedule, setting $t_{\max} = 1000$, with the variance ranging from $1 \times 10^{-4}$ to $2 \times 10^{-2}$. A learned covariance $\Sigma_\theta$ was utilized as described by (Peebles & Xie, 2023). During sampling, 100 steps were used to balance computational efficiency and output quality. The denoising neural network employs a 3-layer encoder-decoder architecture with a hidden layer size of 128. The training process utilized a learning rate of $3 \times 10^{-4}$ with a batch size of 128, followed by a warmup period of 20,000 steps and a linear schedule up to 300,000 steps, with the final learning rate set to $1 \times 10^{-5}$. We implemented LDB using PyTorch 2.3.0 with CUDA 12.1 and Python 3.11. All models were trained and evaluated on 1 NVIDIA A100 GPUs, each with 40GB of memory.

### 5.2 RESULTS ON THE PED DATASET

The experimental results on the PED dataset, as shown in Table 1, highlight LDB's SOTA performance in addressing backmapping challenges. The PED dataset, a benchmark for medium conformational space, was used to evaluate the methods. We sampled each protein structure ten times and reported the mean and standard deviation to ensure robustness.

LDB excels in structural accuracy, outperforming GenZProt and DiAMoNDBack in RMSD for most test proteins. It also achieves significantly lower GED scores, likely due to the internal coordinate representation, which helps maintain valid bond lengths and preserves the original graph structure. This enables LDB to explore a broad conformational space while maintaining fine-grained structural precision, critical for backmapping tasks.

In terms of efficiency and structural validity, LDB consistently delivers superior or competitive results across clash, interaction, and GDR metrics. This demonstrates that LDB not only produces

Table 1: Comparison of structural accuracy and chemical validity on the PED dataset. Our method shows competitive or leading performance in structural accuracy (RMSD, GED) and chemical validity (Clash, Interaction, GDR) metrics.

| | Method | PED00055 | PED00090 | PED00151 | PED00218 |
|---|---|---|---|---|---|
| RMSD ($\downarrow$) | Genzprot | $1.839\pm0.002$ | $2.070\pm0.003$ | $\mathbf{1.629}\pm\mathbf{0.001}$ | $1.800\pm0.002$ |
| | DiAMoNDBack | $1.843\pm0.008$ | $1.958\pm0.014$ | $1.769\pm0.008$ | $1.637\pm0.012$ |
| | Ours | $\mathbf{1.689}\pm\mathbf{0.009}$ | $\mathbf{1.857}\pm\mathbf{0.020}$ | $1.673\pm0.005$ | $\mathbf{1.622}\pm\mathbf{0.015}$ |
| GED ($10^{-2}$; $\downarrow$) | Genzprot | $0.622\pm0.002$ | $1.185\pm0.003$ | $0.678\pm0.002$ | $0.716\pm0.004$ |
| | DiAMoNDBack | $6.683\pm0.024$ | $6.577\pm0.022$ | $1.815\pm0.015$ | $4.385\pm0.032$ |
| | Ours | $\mathbf{0.476}\pm\mathbf{0.004}$ | $\mathbf{0.588}\pm\mathbf{0.004}$ | $\mathbf{0.372}\pm\mathbf{0.003}$ | $\mathbf{0.450}\pm\mathbf{0.004}$ |
| Clash (‰; $\downarrow$) | Genzprot | $0.209\pm0.006$ | $0.390\pm0.014$ | $0.021\pm0.003$ | $1.171\pm0.007$ |
| | DiAMoNDBack | $\mathbf{0.095}\pm\mathbf{0.009}$ | $0.221\pm0.021$ | $\mathbf{0.008}\pm\mathbf{0.002}$ | $1.087\pm0.009$ |
| | Ours | $0.100\pm0.009$ | $\mathbf{0.110}\pm\mathbf{0.013}$ | $0.010\pm0.001$ | $\mathbf{1.080}\pm\mathbf{0.006}$ |
| Interaction ($\downarrow$) | Genzprot | $1.815\pm0.009$ | $1.409\pm0.008$ | $1.605\pm0.015$ | $3.007\pm0.007$ |
| | DiAMoNDBack | $\mathbf{1.613}\pm\mathbf{0.050}$ | $\mathbf{0.945}\pm\mathbf{0.020}$ | $\mathbf{1.468}\pm\mathbf{0.056}$ | $\mathbf{2.627}\pm\mathbf{0.068}$ |
| | Ours | $1.621\pm0.078$ | $0.969\pm0.028$ | $1.485\pm0.051$ | $2.789\pm0.036$ |
| GDR (%; $\downarrow$) | Genzprot | $5.057\pm0.026$ | $7.308\pm0.092$ | $2.173\pm0.017$ | $3.056\pm0.056$ |
| | DiAMoNDBack | $1.700\pm0.101$ | $2.852\pm0.189$ | $0.530\pm0.034$ | $0.928\pm0.076$ |
| | Ours | $\mathbf{1.599}\pm\mathbf{0.080}$ | $\mathbf{1.746}\pm\mathbf{0.145}$ | $\mathbf{0.267}\pm\mathbf{0.028}$ | $\mathbf{0.855}\pm\mathbf{0.056}$ |

accurate structures but also ensures their chemical and physical validity. Its leading clash score indicates fewer unrealistic atomic overlaps, while strong interaction and bond graph accuracy reflect adherence to expected chemical interactions. Although DiAMoNDBack produces reasonable results, its backmapping process approximately **20 times slower** than LDB,, which leverages a latent space approach for efficient structure generation. The robustness of LDB is further confirmed by the generated samples, as seen in Figure 3 and Figure 4.

## 5.3 RESULTS ON THE ATLAS DATASET

The results on the ATLAS dataset, as shown in Table 2 (left half), demonstrate LDB's ability to handle significantly larger and more diverse conformational spaces than PED. Given the extensive variety of proteins in the test set, we selected examples with the best and worst clash loss generated by our method for illustration. The ATLAS dataset includes 15 times more proteins and spans a conformational space 300 times larger than PED, making it a considerably more complex challenge.

Importantly, we did not include DiAMoNDBack in this analysis, as its reproduced results exhibited excessive GED errors. Upon further inspection of the generated structures, we observed frequent graph structure disconnections, likely due to the vast conformational space of the ATLAS dataset, which caused DiAMoNDBack to produce overly diverse and erroneous structures that deviated from the intended targets.

Regarding structural accuracy, LDB consistently outperforms both GenZProt in RMSD across all ATLAS test sets, particularly achieving the lowest RMSD scores in both the overall and worst-case scenarios. This highlights LDB's ability to accurately reconstruct protein structures across a wide range of conformations. The GED results further reinforce this observation, where LDB exhibits significantly lower GED values, indicating its capacity to maintain the correct bond graph structure even in the challenging ATLAS dataset.

In terms of structural validity, LDB also leads in metrics such as Clash and Interaction scores, achieving fewer steric clashes and preserving physical interactions more effectively than the baselines. The consistently lower GDR values across all test cases underline LDB's superior capability in generating chemically valid and physically realistic structures, ensuring that even within larger and more diverse conformational spaces, the model remains robust and reliable.

The visualization of generated samples, as shown in Figure 5, further exemplifies LDB's ability to produce realistic and valid protein structures in challenging conditions. These results substantiate LDB's SOTA performance and validate the effectiveness of our approach in addressing both challenges of large-scale conformational exploration and computational efficiency.

Table 2: Comparison of structural accuracy and chemical validity on the ATLAS and PDB dataset.

| | Method | ATLAS overall | ATLAS best (7jfl_C) | ATLAS worst (7onn_A) | PDB overall | PDB best (T0868) | PDB worst (T0891) |
|---|---|---|---|---|---|---|---|
| RMSD ($\downarrow$) | Genzprot | 1.718±0.157 | 1.484±0.043 | 1.728±0.011 | 1.610±0.162 | 1.318±0.062 | 1.764±0.073 |
| | DiAMoNDBack | - | - | - | 1.294±0.192 | 1.120±0.065 | 1.391±0.053 |
| | Ours | **1.539±0.176** | **1.435±0.043** | **1.396±0.022** | **1.236±0.183** | **1.106±0.079** | **1.309±0.089** |
| GED ($10^{-2}$; $\downarrow$) | Genzprot | 0.715±0.188 | 0.441±0.012 | 0.920±0.026 | 0.382±0.219 | 0.177±0.034 | 0.393±0.058 |
| | DiAMoNDBack | - | - | - | 0.714±0.003 | 0.454±0.001 | 0.708±0.001 |
| | Ours | **0.391±0.044** | **0.306±0.011** | **0.495±0.010** | **0.162±0.093** | **0.083±0.006** | **0.241±0.035** |
| Clash (‰; $\downarrow$) | Genzprot | 0.232±0.265 | 0.060±0.039 | **0.294±0.245** | 0.660±1.123 | 0.046±0.043 | 3.602±0.076 |
| | DiAMoNDBack | - | - | - | **0.422±0.905** | 0.005±0.010 | **3.435±0.000** |
| | Ours | **0.047±0.096** | **0.009±0.013** | 0.642±0.478 | 0.435±0.907 | **0.000** | 3.461±0.020 |
| Interaction ($\downarrow$) | Genzprot | 1.627±0.346 | 1.042±0.123 | 1.589±0.041 | 1.577±0.708 | 0.745±0.087 | 2.114±0.141 |
| | DiAMoNDBack | - | - | - | 1.027±0.683 | 0.456±0.137 | 1.016±0.247 |
| | Ours | **1.128±0.329** | **0.764±0.212** | **1.002±0.049** | **0.843±0.623** | **0.322±0.106** | **0.692±0.182** |
| GDR (%; $\downarrow$) | Genzprot | 4.140±1.505 | 1.274±0.273 | 5.111±0.332 | 3.480±1.330 | 1.037±0.375 | 2.525±0.833 |
| | DiAMoNDBack | - | - | - | 0.918±0.360 | 0.438±0.229 | 0.609±0.215 |
| | Ours | **0.926±0.391** | **0.279±0.117** | **0.920±0.065** | **0.533±0.355** | **0.046±0.056** | **0.245±0.247** |

## 5.4 RESULTS ON THE PDB DATASET

The results on the PDB dataset, as shown in Table 2 (right half), demonstrate LDB's robustness in handling large static datasets with over 60,000 single-conformation proteins—700 times more than PED. Unlike dynamic datasets such as ATLAS and PED, PDB contains steady-state structures without molecular dynamics data, posing the challenge of reconstructing static structures in the absence of conformational diversity.

LDB achieves superior or competitive results in RMSD and GED compared to GenZProt and Di-AMoNDBack, particularly excelling in overall RMSD and both best- and worst-case structures. This highlights LDB's consistent ability to reconstruct high-fidelity structures, even without conformational diversity. GED results further confirm the model's ability to maintain structural integrity across a wide range of protein types.

In terms of structural validity, LDB outperforms the baselines in Clash and GDR scores, ensuring both accuracy and physical plausibility. LDB also achieves the highest Interaction scores, preserving critical atomic interactions essential for functional analysis. These results confirm LDB's capability in generating chemically valid, physically realistic steady-state structures.

Figure 6 visually illustrates LDB's effectiveness, showcasing its ability to produce structurally sound results on real-world protein data. Overall, LDB demonstrates consistent superiority in accuracy (RMSD, GED) and structural validity (Interaction, GDR), without sacrificing inference efficiency, making it well-suited for large-scale applications in protein modeling and drug discovery.

## 5.5 ABLATION STUDIES

To evaluate the contributions of key model components, we conducted ablation studies on the PED dataset, comparing our discrete latent space approach (VQ-VAE+diffusion) with two alternatives: a continuous latent space model (VAE+diffusion) and a flow-based variant (VQ-VAE+flow). Flow matching, known for balancing stochasticity and structure in recent tasks, offers efficient probability flows (Irwin et al., 2024; Jing et al., 2024), but for protein backmapping tasks with large conformational spaces, diffusion's ability to explore diverse conformations proves more effective. Each component plays a distinct role in improving structural accuracy and validity.

Firstly, our discrete latent space (VQ-VAE) shows clear advantages over the continuous VAE-based method. By discretizing the latent space, our model can better preserve the bond graph consistency, which is crucial for maintaining accurate internal structures. This is reflected in the significantly lower GED scores as shown in Table 3. The discrete space effectively reduces errors related to bond lengths and angles, which leads to better structural precision.

Secondly, the diffusion process proves superior to the flow-based approach (VQ-VAE+flow) in handling large conformational spaces. Diffusion leverages stochastic noise, which allows for exploration across diverse conformations while maintaining structure validity. This is evident in the

Table 3: Ablation on PED dataset for the model architecture.

| | Method | PED00055 | PED00090 | PED00151 | PED00218 |
|---|---|---|---|---|---|
| RMSD ($\downarrow$) | VAE+diffusion | $1.786_{\pm0.007}$ | $1.938_{\pm0.016}$ | $1.820_{\pm0.013}$ | $1.706_{\pm0.018}$ |
| | VQ-VAE+flow | $1.794_{\pm0.008}$ | $1.918_{\pm0.015}$ | $1.838_{\pm0.011}$ | $1.674_{\pm0.013}$ |
| | VQ-VAE+diffusion | $\mathbf{1.689}_{\pm\mathbf{0.009}}$ | $\mathbf{1.857}_{\pm\mathbf{0.020}}$ | $\mathbf{1.673}_{\pm\mathbf{0.005}}$ | $\mathbf{1.622}_{\pm\mathbf{0.015}}$ |
| GED ($10^{-2}$; $\downarrow$) | VAE+diffusion | $1.033_{\pm0.005}$ | $1.210_{\pm0.008}$ | $0.378_{\pm0.003}$ | $0.898_{\pm0.020}$ |
| | VQ-VAE+flow | $0.504_{\pm0.004}$ | $\mathbf{0.583}_{\pm\mathbf{0.005}}$ | $0.405_{\pm0.005}$ | $0.495_{\pm0.014}$ |
| | VQ-VAE+diffusion | $\mathbf{0.476}_{\pm\mathbf{0.004}}$ | $0.588_{\pm0.004}$ | $\mathbf{0.372}_{\pm\mathbf{0.003}}$ | $\mathbf{0.450}_{\pm\mathbf{0.004}}$ |
| Clash (‰; $\downarrow$) | VAE+diffusion | $0.120_{\pm0.014}$ | $0.212_{\pm0.019}$ | $0.019_{\pm0.005}$ | $1.154_{\pm0.015}$ |
| | VQ-VAE+flow | $0.103_{\pm0.015}$ | $0.228_{\pm0.029}$ | $0.020_{\pm0.005}$ | $1.118_{\pm0.006}$ |
| | VQ-VAE+diffusion | $\mathbf{0.100}_{\pm\mathbf{0.009}}$ | $\mathbf{0.110}_{\pm\mathbf{0.013}}$ | $\mathbf{0.010}_{\pm\mathbf{0.001}}$ | $\mathbf{1.080}_{\pm\mathbf{0.006}}$ |
| Interaction ($\downarrow$) | VAE+diffusion | $1.496_{\pm0.043}$ | $1.068_{\pm0.037}$ | $1.512_{\pm0.066}$ | $2.802_{\pm0.061}$ |
| | VQ-VAE+flow | $\mathbf{1.423}_{\pm\mathbf{0.080}}$ | $1.113_{\pm0.052}$ | $1.547_{\pm0.055}$ | $\mathbf{2.763}_{\pm\mathbf{0.057}}$ |
| | VQ-VAE+diffusion | $1.621_{\pm0.078}$ | $\mathbf{0.969}_{\pm\mathbf{0.028}}$ | $\mathbf{1.485}_{\pm\mathbf{0.051}}$ | $2.789_{\pm0.036}$ |
| GDR (%; $\downarrow$) | VAE+diffusion | $2.689_{\pm0.118}$ | $3.726_{\pm0.207}$ | $0.397_{\pm0.033}$ | $1.994_{\pm0.211}$ |
| | VQ-VAE+flow | $1.890_{\pm0.107}$ | $2.823_{\pm0.286}$ | $0.406_{\pm0.039}$ | $1.426_{\pm0.052}$ |
| | VQ-VAE+diffusion | $\mathbf{1.599}_{\pm\mathbf{0.080}}$ | $\mathbf{1.746}_{\pm\mathbf{0.145}}$ | $\mathbf{0.267}_{\pm\mathbf{0.028}}$ | $\mathbf{0.855}_{\pm\mathbf{0.056}}$ |

RMSD and Clash metrics, where our diffusion-based model consistently achieves better results. Specifically, the diffusion process allows for finer adjustments during multi-step denoising, leading to fewer steric clashes and better interaction preservation, as indicated by lower Clash and GDR scores.

These results highlight the complementary strengths of discrete latent space for preserving fine structural details and diffusion for maintaining structural validity across diverse conformations. Combining these components enhances both accuracy and efficiency in protein backmapping, making our approach robust and effective for large conformational spaces.

## 6 CONCLUSION

In this paper, we introduced LDB, a denoising diffusion backmapping method operating in latent space. By implicitly incorporating equivariance and internal coordinates into a discrete low-dimensional node-level latent representation, we effectively preserved structural information while simplifying the diffusion process, thereby enhancing both efficiency and performance. This method addresses the inefficiencies and accuracy challenges of direct diffusion in coordinate space, as well as the difficulties in learning simple prior distributions that struggle to capture diverse conformational spaces. Our experiments demonstrate that LDB achieves SOTA accuracy across various datasets while maintaining higher structural validity. For future work, we aim to extend this framework to model continuous time trajectories, which will allow better prediction of dynamic protein behaviors. Additionally, this versatile framework can be adapted for other tasks in protein design and beyond.

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

# A APPENDIX

## A.1 DATASET PREPROCCESS

**PED:** The PED contains structural ensembles of various proteins, including numerous intrinsically disordered proteins (IDPs). In line with the approach taken by the GenZProt model (Yang & Gómez-Bombarelli, 2023), we initially selected 88 proteins from the PED dataset. To ensure compatibility with prior work, we further filtered out three proteins—PED00125e000, PED00126e000, and PED00161e002—that contain non-canonical amino acids, following the methodology of DiAMoNDBAck. This left us with a total of 85 proteins for training. For evaluation purposes, we used the same test set as previous studies, consisting of four PED proteins: PED00151ecut0, PED00090e000, PED00055e000, and PED00218e000, which contain 20 to 140 frames, and the remaining proteins were used for training.

**ATLAS:** The ATLAS dataset consists of all-atom molecular dynamics (MD) simulations for 1,390 non-membrane proteins, each chosen to represent all eligible ECOD structural classes (Schaeffer et al., 2017). For each protein, three replicate simulations of 100 ns are provided, with each simulation containing 10,000 frames. Following the preprocessing steps used in the Alphaflow framework (Jing et al., 2024), 300 conformations per protein were randomly sampled for training. To maintain consistency in our experment, we excluded 95 sequences with lengths greater than 512 residues. The final test set was composed of proteins whose corresponding PDB entries were deposited after May 1, 2019.

**PDB:** The PDB dataset comprises protein structures from the Protein Data Bank (PDB), collated in the SidechainNet extension of ProteinNet. In accordance with the preprocessing strategy used by DiAMoNDBAck (Jones et al., 2023), we filtered out sequences with incomplete side-chain coordinates for non-terminal residues, as well as configurations with $C\alpha$-$C\alpha$ distances outside the 2.7-4.1 Å range. Additionally, we removed sequences containing four or more disconnected chains and those with fewer than five residues. After these steps, we retained 65,360 structures for training. Finally, we further refined the dataset by excluding 3,270 structures with sequence lengths greater than 512 residues, ensuring a robust dataset for our experiments.

## A.2 EVALUATION METRICS

**Root Mean Squared Distance (RMSD):** The RMSD calculates the average distance between corresponding atoms in two structures, effectively quantifying the difference between the reference and generated structures, thereby assessing the quality of the reconstruction. For a generated structure $x^{\text{gen}}$ and reference structure $x^{\text{ref}}$, RMSD is computed as:

$$\text{RMSD} = \sqrt{\frac{1}{n} \sum_{i=1}^{n} \|x_i^{\text{gen}} - x_i^{\text{ref}}\|^2},$$

where $n$ is the total number of atoms. This metric provides a direct assessment of reconstruction quality, with lower values indicating closer alignment to the reference structure.

**Graph Edit Distance (GED):** The quality of generated samples is assessed based on how well they retain the original chemical bond graph structure, quantified by the graph edit distance ratio $\lambda(G^{\text{gen}}, G^{\text{ref}})$ between the generated graph and the reference graph. Given the generated structure $x^{\text{gen}}$ and reference structure $x^{\text{ref}}$, and their respective edge lists edge_list, the graph loss is calculated as:

$$\text{GED} = \frac{1}{e} \sum_{(i,j) \in \text{edge\_list}} \left( \|x_i^{\text{gen}} - x_j^{\text{gen}}\| - \|x_i^{\text{ref}} - x_j^{\text{ref}}\| \right)^2,$$

where $e$ is the number of edges. This metric evaluates the structural fidelity of the generated bond graph relative to the reference.

**Steric Clash Score:** The generated structure should have a reasonable atomic distribution. We report the ratio of steric clashes among all atom-atom pairs, where a distance smaller than 1.2 Å between any two atoms is considered a steric clash (Yang & Gómez-Bombarelli, 2023; Jones et al., 2023). For a generated structure $x^{\text{gen}}$, the score is calculated by identifying all atom pairs within a

distance smaller than 1.2 Å. The ratio of steric clashes is defined as:

$$\text{Clash Score} = \frac{\text{Number of clashes in } x^{\text{gen}}}{\text{Total number of atom pairs in } x^{\text{gen}}}.$$

**Interaction Score:** We define the Interaction Score as a single value to evaluate the physical plausibility of the generated structures. This score captures two types of interactions: (1) hydrogen bonds, ion-ion interactions, and dipole-dipole interactions between atom pairs within 3.3 Å; and (2) $\pi$-$\pi$ stacking interactions among aromatic ring pairs (PHE, TYR, TRP, HIS) with center distances smaller than 5.5 Å. The Interaction Score is computed as:

$$L = \sum_{(x,y)\in\mathcal{A}} \max(\|x - y\|_2^2 - 4.0, 0.0) + \sum_{(x,y)\in\mathcal{P}} \max(\|x - y\|_2^2 - 6.0, 0.0)$$

where $\mathcal{A}$ represents interacting atom pairs and $\mathcal{P}$ represents pairs of aromatic rings. Lower interaction scores indicate more chemically realistic structures.

**Graph Difference Ratio (GDR):** The GDR measures the fidelity of generated bond graphs compared to reference bond graphs, which are constructed based on covalent bond distances. A bond is defined between two atoms if their distance is smaller than a threshold, calculated as the sum of their covalent radii scaled by a factor of 1.3 to account for permissible bond length variations:

$$G_{ij} = \begin{cases} 1 & \text{if } \|\mathbf{x}_i - \mathbf{x}_j\| < (\text{radius}_i + \text{radius}_j) \times \text{scale}, \\ 0 & \text{otherwise.} \end{cases}$$

The GDR is then calculated as:

$$\text{GDR} = \frac{\|G_{\text{true}} - G_{\text{gen}}\|_1}{\|G_{\text{true}}\|_1},$$

where $G_{\text{true}}$ and $G_{\text{gen}}$ are the reference and generated bond graphs, respectively. Lower GDR values indicate better structural fidelity.

### A.3 Representation for Protein Structure

Protein structure representations are essential for tasks such as protein design, folding prediction, and structural backmapping. Numerous approaches have been developed to represent protein structures in a computationally efficient manner. Below, we discuss several common methods of representation.

**Voxel Representation** Voxel representations divide 3D space into a grid, where each voxel indicates the presence or absence of atoms. This method provides a clear way to capture spatial information, but it can be computationally demanding due to the high dimensionality of the voxel grid, especially when applied to large macromolecules. It is mainly utilized in tasks that require explicit spatial reasoning, such as molecular docking simulations. Several studies (Masuda et al., 2020; Stieffenhofer et al., 2020; 2021; Shmilovich et al., 2022) have implemented atomic density grids, allowing for the entire molecule to be generated in one step by producing a density over the voxelized 3D space. However, these grids lack the desirable property of equivariance and often necessitate separate fitting algorithms, which adds complexity to the modeling process.

**Coordinate Representation** Coordinate representation captures the precise spatial arrangement of each atom in a protein using Cartesian coordinates, making it a standard approach in many molecular modeling techniques. This method effectively preserves the geometric properties of protein structures, facilitating accurate modeling tasks. However, directly integrating Cartesian coordinates into deep learning models presents challenges, particularly the need for translational and rotational invariance, which necessitates specific constraints within the network. Furthermore, the high dimensionality of coordinate data increases computational complexity, especially in large-scale datasets, while uneven data distribution can impede learning efficiency. Consequently, advanced learning strategies are often required to address these challenges (Hoogeboom et al., 2022; Wu et al., 2022).

**Internal Coordinate Representation**  The internal coordinate representation utilizes bond lengths, bond angles, and dihedral angles to reduce the degrees of freedom compared to Cartesian coordinates, resulting in a more compact and efficient representation (Jing et al., 2022; Eguchi et al., 2022). This approach inherently encodes the geometric constraints of molecular structures, enhancing computational efficiency while eliminating redundant spatial information. It is particularly well-suited for backmapping tasks, where known reference points facilitate the reconstruction of full-atom coordinates. By relying on internal coordinates, the process conforms to the physical and chemical constraints of the system, enabling the accurate and efficient generation of all-atom structures.

**Latent Representation**  Latent diffusion models have demonstrated significant success across various generative tasks, including image (Vahdat et al., 2021), point cloud (Zeng et al., 2022), text (Li et al., 2022), audio (Liu et al., 2023), and molecular generation (Xu et al., 2023). In the context of protein structures, latent representations offer a compact and efficient method for modeling by embedding them into lower-dimensional spaces, thereby simplifying both the generation and design processes. (Xu et al., 2023) introduced a geometric latent diffusion model for 3D molecular generation that ensures roto-translational equivariance within the latent space, enhancing the modeling of small molecular geometries. (Fu et al., 2024) proposed a latent diffusion model that adeptly captures protein geometry, facilitating the efficient generation of novel protein backbones through latent node and edge features. Similarly, (Hayes et al., 2024) employed latent space modeling to simulate protein evolution, showcasing its capability to co-design protein sequences and structures. Collectively, these methods reduce computational complexity while preserving high-quality protein generation and designability.

## A.4 BASELINE MODELS

**DiAMoNDBack** reconstructs full-atom structures from CG representations by directly performing diffusion on atomic coordinates. The method introduces Gaussian noise to atomic coordinates in a forward diffusion process, transforming the data into a noise distribution. During inference, the reverse process iteratively removes the noise to recover the original coordinates. To ensure invariance to global rotations and translations, each residue is represented in a canonical reference frame defined by its neighboring residues.

The model employs a U-Net-based denoising network to predict clean coordinates at each diffusion step. It follows an autoregressive approach, generating the structure residue by residue, starting from the N-terminus. Each residue is predicted conditionally based on the previously generated residues, ensuring both local accuracy and global consistency.

DiAMoNDBack focuses on directly modeling in Cartesian coordinate space, allowing it to generate diverse conformations while maintaining structural integrity. However, its autoregressive nature significantly increases inference time compared to non-autoregressive models.

**GenZProt** reconstructs full-atom structures from CG representations using a VAE framework. Instead of predicting Cartesian coordinates directly, it predicts internal coordinates, including bond lengths, bond angles, and torsion angles, ensuring chemical and physical validity while avoiding steric clashes.

The model uses a hierarchical architecture to process input data. The encoder captures geometric information at three levels: atomic interactions within 9 Å, residue-level interactions, and long-range residue interactions within 21 Å. The decoder generates internal coordinates for each residue, which are converted to Cartesian coordinates using a Z-matrix formulation. This representation reduces the complexity of direct Cartesian prediction while preserving physical constraints.

Training is guided by physics-informed loss functions, focusing on bond lengths, bond angles, torsion angles, and steric clash avoidance. GenZProt learns a prior distribution of protein structures in the latent space, capturing the most plausible conformations for a given CG representation. This allows it to efficiently reconstruct full-atom structures with high chemical accuracy.

A.5    SAMPLE STRUCTURE

**PED00055**

(a) Ground truth        (b) LDB        (c) Genzprot        (d) DiAMoNDBack

**PED00090**

(a) Ground truth        (b) LDB        (c) Genzprot        (d) DiAMoNDBack

**PED00151**

(a) Ground truth        (b) LDB        (c) Genzprot        (d) DiAMoNDBack

**PED00218**

(a) Ground truth        (b) LDB        (c) Genzprot        (d) DiAMoNDBack

Figure 4: Comparison of sample protein structures across different methods. The structures shown are from PED. Each row represents a different protein (PED00055, PED00090, PED00151, and PED00218), our method remain close to the reference conformation and maintain good structural integrity

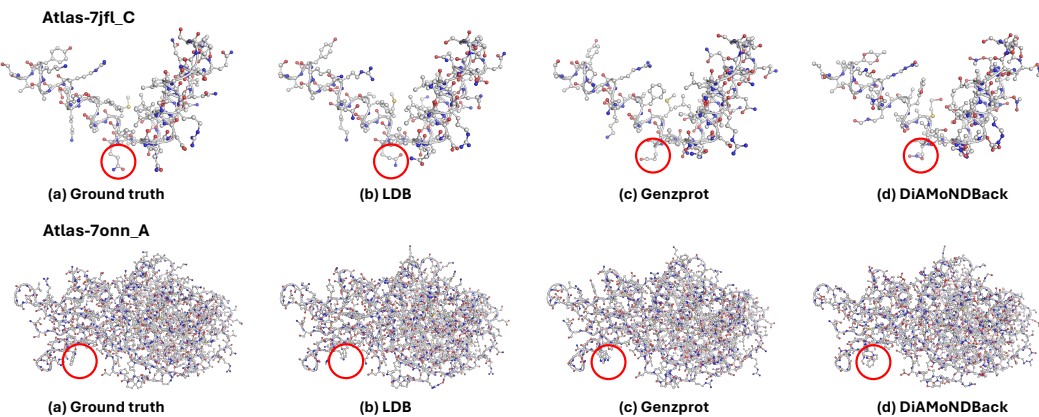

Figure 5: Visualization of protein structure generation from the ATLAS dataset. Our method remain close to the reference conformation and maintain good structural integrity

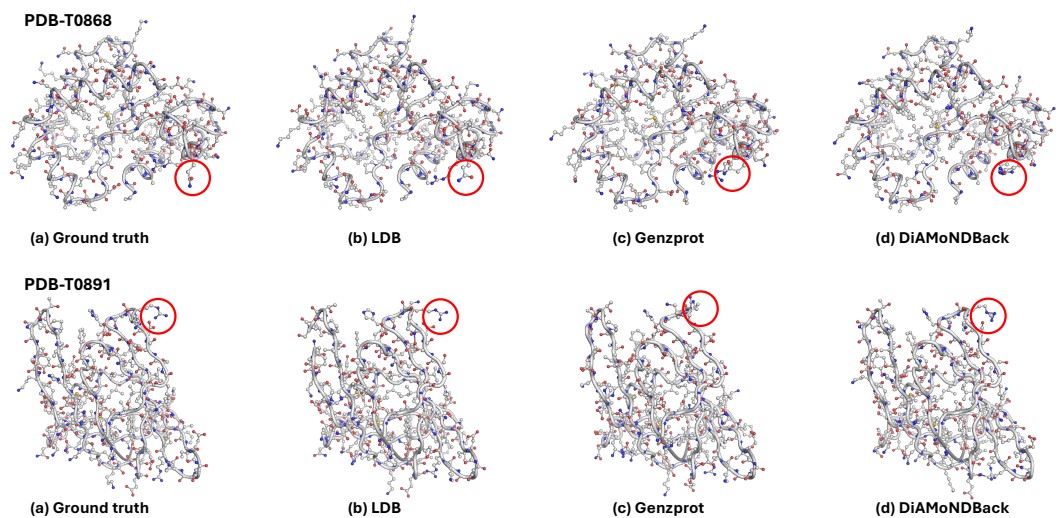

Figure 6: Visualization of protein structure generation from the PDB dataset. Our method remain close to the reference conformation and maintain good structural integrity

## A.6 ADDITIONAL EXPERIMENTAL RESULTS

### A.6.1 STRUCTURE DIVERSITY

We evaluated the diversity of the generated protein structures, acknowledging that while a single CG model typically corresponds to a unique all-atom structure in practical applications, the inherent information compression in CG representations often allows one CG model to correspond to multiple plausible all-atom structures. As a result, the model must learn a distribution of possible mappings, making it important to balance structural fidelity with diversity in the generated output.

The diversity metric quantifies whether the generated structures exhibit meaningful variation while maintaining consistency with the reference structure. To assess this, we adopted the Diversity Score from the Diamondback framework. This score compares the structural error between the backmapped structures ($RMSD_{gen}$) and the reference structure ($RMSD_{ref}$) with the structural variability among the backmapped structures themselves ($RMSD_{gen}$). The Diversity Score is computed as follows:

Table 4: Diversity scores of generated structures on the PED dataset.

| | Method | PED00055 | PED00090 | PED00151 | PED00218 |
|---|---|---|---|---|---|
| | Genzprot | $0.909 \pm 0.001$ | $0.903 \pm 0.001$ | $0.888 \pm 0.001$ | $0.893 \pm 0.001$ |
| Diversity ($\downarrow$) | DiAMoNDBack | $0.466 \pm 0.022$ | $0.479 \pm 0.014$ | $\mathbf{0.423 \pm 0.002}$ | $0.484 \pm 0.001$ |
| | Ours | $\mathbf{0.447 \pm 0.001}$ | $\mathbf{0.465 \pm 0.001}$ | $0.424 \pm 0.001$ | $\mathbf{0.480 \pm 0.005}$ |

Table 5: Jensen-Shannon Divergence of torsion angle distributions on the PED dataset.

| | Method | PED00055 | PED00090 | PED00151 | PED00218 |
|---|---|---|---|---|---|
| | Genzprot | $0.326 \pm 0.149$ | $0.307 \pm 0.111$ | $0.177 \pm 0.074$ | $0.470 \pm 0.140$ |
| JSD ($\downarrow$) | DiAMoNDBack | $0.241 \pm 0.134$ | $0.163 \pm 0.078$ | $\mathbf{0.041 \pm 0.017}$ | $\mathbf{0.194 \pm 0.100}$ |
| | Ours | $\mathbf{0.193 \pm 0.091}$ | $\mathbf{0.147 \pm 0.052}$ | $0.047 \pm 0.023$ | $0.239 \pm 0.147$ |

$$\text{RMSD}_{\text{ref}} = \frac{1}{G} \sum_{i=1}^{G} \text{RMSD}(\mathbf{x}_i^{\text{gen}}, \mathbf{x}_i^{\text{ref}})$$

$$\text{RMSD}_{\text{gen}} = \frac{2}{G(G-1)} \sum_{i=1}^{G} \sum_{j<i} \text{RMSD}(\mathbf{x}_i^{\text{gen}}, \mathbf{x}_j^{\text{gen}})$$

$$\text{DIV} = 1 - \frac{\text{RMSD}_{\text{gen}}}{\text{RMSD}_{\text{ref}}}$$

Here, $G$ represents the number of all-atom structures generated from a single CG model, $\mathbf{x}^{\text{gen}}$ denotes the predicted structure coordinates, and $\mathbf{x}^{\text{ref}}$ represents the reference structure coordinates.

As shown in Table 4, the Diversity Score provides insight into the variability of the generated structures relative to the reference. It is important to note, however, that higher diversity does not necessarily indicate better backmapping performance. The primary objective remains to achieve a close fit to the reference structure while allowing for a reasonable degree of diversity to reflect the distribution of plausible all-atom conformations.

### A.6.2 TORSION ANGLE DISTRIBUTION

Torsion angles, particularly chi1 ($\chi_1$) angles, are highly prevalent and exhibit greater degrees of freedom in protein residues. To analyze the distribution of $\chi_1$ angles, we excluded residues such as Gly and Ala, which lack $\chi_1$ angles. Using kernel density estimation (KDE), we visualized the $\chi_1$ angle distributions and quantified the differences between the predicted structures and reference structures by calculating the Jensen-Shannon divergence (JSD). The results are summarized in Table 5.

To further evaluate the model's performance, we visualized the $\chi_1$ angle distributions for all residues in the first test protein, PED00055. As shown in Figure 7, our model closely aligns with the reference distribution for most residues, with no significant deviations observed. Additionally, we selected several representative residues from this protein and visualized their fitted $\chi_1$ distributions in Figure 8. These visualizations demonstrate the model's ability to capture the multi-modal nature of the reference distributions, accurately reflecting the inherent variability of torsion angles in protein structures.

### A.7 INTERNAL COORDINATE SYSTEM

Internal coordinates are computed for each residue to describe the geometric and chemical relationships among its atoms. These coordinates include bond lengths, bond angles, and dihedral angles, capturing the spatial arrangement of up to 13 heavy atoms (excluding the central $C_\alpha$ atom). Definition of Internal Coordinates:

**Bond Lengths:** Bond lengths represent the distances between two bonded atoms. For two atoms $i$ and $j$, the bond length $d_{ij}$ is:

$$d_{ij} = \|\mathbf{x}_i - \mathbf{x}_j\|,$$

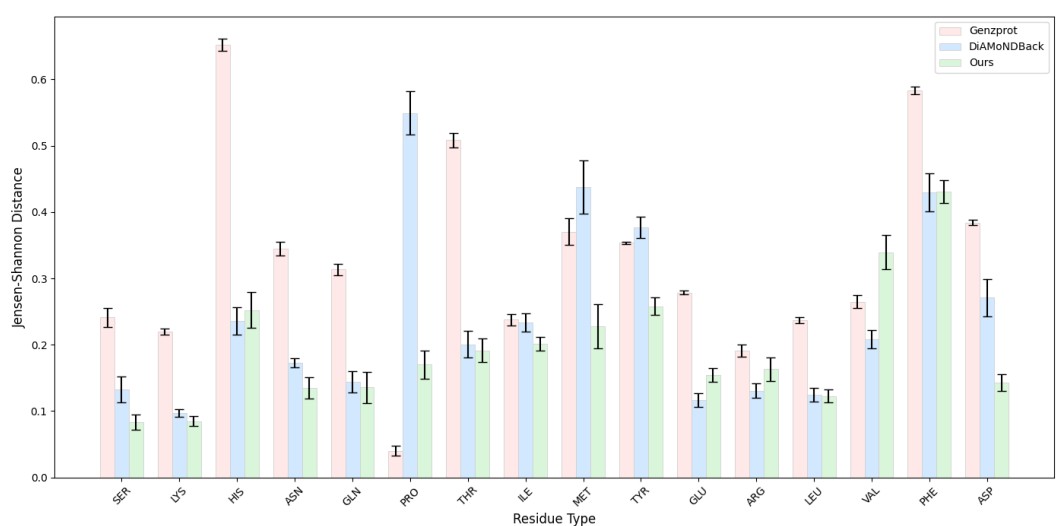

Figure 7: Residue-Wise Divergence for Protein PED00055. JSD of torsion angle distributions for each residue in protein PED00055. Our method demonstrates better similarity to the reference distribution compared to other methods, with no extreme outliers observed.

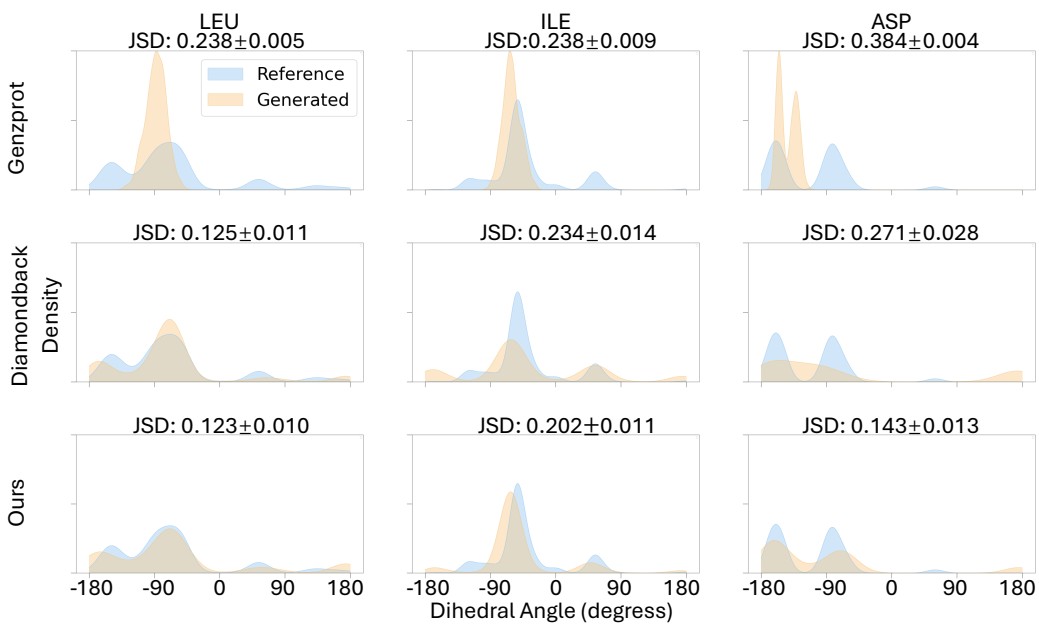

Figure 8: Torsion Angle Distributions for Selected Residues. KDE of torsion angle distributions for LEU, ILE, and ASP residues. Our method achieves better alignment with the reference distribution and successfully captures the multi-modal nature of the torsion angle distributions.

where $\mathbf{x}_i$ and $\mathbf{x}_j$ are their Cartesian coordinates.

**Bond Angles:** Bond angles describe the angles formed by three consecutive atoms. For atoms $i$, $j$, and $k$, the bond angle $\theta_{ijk}$ is calculated as:

$$\theta_{ijk} = \arccos\left(\frac{(\mathbf{x}_i - \mathbf{x}_j) \cdot (\mathbf{x}_k - \mathbf{x}_j)}{\|\mathbf{x}_i - \mathbf{x}_j\|\|\mathbf{x}_k - \mathbf{x}_j\|}\right),$$

ensuring the spatial orientation of bonded atoms.

**Dihedral Angles:** Dihedral angles measure the rotation around a bond and are defined by four consecutive atoms. For atoms $i$, $j$, $k$, and $l$, the dihedral angle $\tau_{ijkl}$ is:

$$\tau_{ijkl} = \arctan 2\left(\frac{(\mathbf{b}_1 \times \mathbf{b}_2) \cdot \mathbf{b}_3}{\|\mathbf{b}_2\|\mathbf{b}_1 \cdot \mathbf{b}_3}, (\mathbf{b}_1 \times \mathbf{b}_2) \cdot (\mathbf{b}_2 \times \mathbf{b}_3)\right),$$

where:

$$\mathbf{b}_1 = \mathbf{x}_j - \mathbf{x}_i, \quad \mathbf{b}_2 = \mathbf{x}_k - \mathbf{x}_j, \quad \mathbf{b}_3 = \mathbf{x}_l - \mathbf{x}_k.$$

Dihedral angles are critical for capturing the rotational flexibility of residues, particularly in side chains.

The described methodology is applied to convert protein structures into internal coordinates in two stages, following a predefined processing order. Typically, the backbone atoms are processed first to establish the structural framework, which is then used as a reference for the sequential conversion of side chain atoms.

**Backbone Atoms:** First, the backbone atoms of each residue are converted into internal coordinates using the $C_\alpha$ atoms of the previous, current, and next residues.

**Sidechain Atoms:** Once the backbone coordinates are reconstructed, the side chain atoms are converted. Each residue starts with known backbone atoms (N, $C_\alpha$, C ), which serve as references. Using these references, the side chain atoms are sequentially converted.

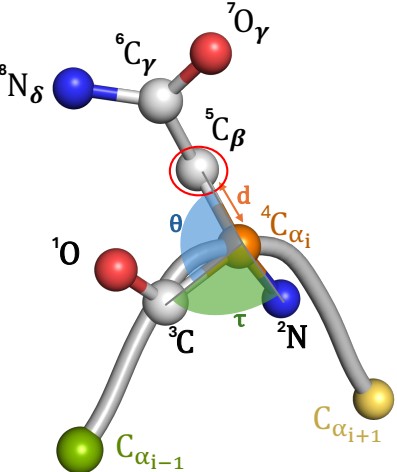

Figure 9: Schematic representation of internal coordinates in a protein residue

Figure 9 provides a schematic representation of the backbone and side chain atoms of a residue, highlighting the internal coordinate framework. For illustrative purposes, the conversion process is demonstrated using the $C_\beta$ atom (labeled as atom 5) as an example:

1. The bond length $d$ is computed as the distance between $C_\beta$ (atom 5) and C (atom 4).

2. The bond angle $\theta$ is calculated as the angle formed by N (atom 3), C (atom 4), and $C_\beta$ (atom 5).

      3. The dihedral angle $\tau$ is determined from the planes formed by $C_\alpha$ (atom 2), N (atom 3), C (atom 4), and $C_\beta$ (atom 5).

This systematic process ensures that all atoms, including both backbone and side chain atoms, are represented in a consistent and compact internal coordinate framework.

## A.8 ABLATION MODEL DETAILS

**VAE+diffusion**: This ablation model removes the VQ component and directly employs diffusion in a continuous latent space. The input to the diffusion model is a $N \times 36$ continuous latent representation, where $N$ is the protein sequence length, and 36 corresponds to the dimensionality of the continuous latent embedding for each residue.

**VQVAE+flow**: This ablation replaces the diffusion process with a flow-matching approach, which interpolates between a noise-injected source and the low-dimensional discrete latent representation. The flow matching framework learns a conditional vector field $u_t$ to align interpolated states $x_t$ with the target latent representation $x_1$ at different time steps $t$. Specifically, the interpolation is defined as:

$$x_t = (1 - t)x_0 + tx_1,$$

where $x_0$ is the noise and $x_1$ is the discrete latent representation. The conditional vector field is learned to satisfy:

$$u_t = \frac{x_1 - (1 - t)x_t}{t},$$

allowing the model to progressively refine the interpolated states toward the target representation. During inference, we use the `dopri5` solver to integrate the learned vector field.

