# OpenReview forum: "The Latent Road to Atoms: Backmapping Coarse-grained Protein Structures with Latent Diffusion"
_ICLR.cc/2025/Conference — Submitted to ICLR 2025_

### Official Review · Reviewer_25tp · 2024-10-19

**Soundness:** 1
**Presentation:** 1
**Contribution:** 2
**Rating:** 3
**Confidence:** 4

**Summary:**

Latent Diffusion Backmapping (LDB) is a generative model approach for modeling protein conformations. A discrete latent space is learned over all-atom protein structures with a VQ-VAE followed by a diffusion model to sample protein conformations. The work claims state-of-the-art (SOTA) results on three protein datasets which I do not agree with. LDB does not achieve SOTA on all metrics and the metrics are confusing as to whether they are relevant to evaluating protein ensembles.

**Strengths:**

* Discrete latent diffusion approach is novel to modeling protein conformations.
* LDB outperforms Genzprot and DiAMoNDBack on most metrics across the three datasets.

**Weaknesses:**

I have several major concerns.

* After having read the paper twice (including the appendix), I still do not understand the method. The problem definition in section 3 is to sample from p(x|X,V) where x is the coordinates and (X, V) is the coarse grained (CG) representation. Then the method in section 4 proceeds to describe a VQ-VAE for compressing protein structures then a graph latent diffusion model to sample discrete latents. Which part is sampling x and which part is sampling (X, V)? The methods section is poorly written with many details left unanswered.

* The paper claims several times that "diffusion in high-dimensional spaces complicates the multi-step denoising process" but provides no evidence of this. In fact, if the latent dimensionality is 36 then isn't this higher dimension than the 3 dimensional coordinates? I would have liked to see a direct comparison or some references but none are provided.

* I do not understand the metrics. If the benchmarks are protein conformation datasets then there should be distributional metrics such as in AlphaFlow [1]. For instance, what does it mean for computing RMSD to PED00055 when it has 20-140? Do you take the min RMSD of the model's samples to the frames? Each metric described in the appendix lacks details on how to calculate them.

* The paper in its current form is not reproducible. There just lacks too much details on the model, metrics, baselines and ablations (how was VAE-difusion and VQ-VAE-flow implemented?). Why was AlphaFlow not compared to?


[1] https://arxiv.org/abs/2402.04845

**Questions:**

My major questions are above. Additional questions.

* Line 168. What is n_f? What are "element types of the protein"? Why are the amino acids real valued instead of integer or categorical?

* Line 177. There are no details with how the internal coordinates are obtained. This is another reason why this work is not reproducible due to the lack of algorithmic details.

* Line 185. The work claims to generate all-atom structures but then this states only the C-alpha coordinates are used. So which is it?

* Line 253. The authors point to Jing et al 2022 to reconstruct full-atom structure but Jing et al 2022 is torsional diffusion for small molecule conformations. How does this relate to the protein structures here?

* Line 273. Where is your evidence of 3D diffusion doing poorly? We know SOTA methods like AlphaFold3 [1] and RFdiffusion [2] work very well in 3D.

* Why was AlphaFlow not compared to?

[1] https://www.nature.com/articles/s41586-024-07487-w

[2] https://www.nature.com/articles/s41586-023-06415-8

---

> ### Author Response · Authors · 2024-11-22
>
> Thank you for your thorough review and valuable feedback. Below, we address each of the raised concerns in detail.
>
> ---
>
> 1.**Distinction Between Backmapping and Protein Conformation/Ensemble Generation Tasks**
>
> We would like to first clarify the fundamental differences between *backmapping* and *protein conformation/ensemble generation* tasks, as they address distinct objectives in protein modeling.
>
> Backmapping focuses on reconstructing the original all-atom structure from a coarse-grained (CG) representation, prioritizing accuracy and fidelity to the reference structure. In contrast, protein conformation or ensemble generation involves creating a diverse set of plausible protein conformations based on a given sequence or additional guiding information, with an emphasis on capturing structural variability and diversity.
>
> While both tasks are essential for understanding protein structure and dynamics, backmapping is primarily concerned with precise atomic-level recovery from simplified representations, whereas conformation generation serves to explore the conformational space of proteins. This distinction is critical to understanding the unique challenges and evaluation criteria associated with each task.
>
> ---
>
> 2.**On Sampling Atomic Coordinates and Discrete Representations**
>
> Regarding the question about why we sample both atomic coordinates and discrete representations, we define backmapping as the process of reconstructing all-atom coordinates from a given CG model. To make this process more efficient and reduce learning complexity, we divide it into two stages:
>
> - **Stage 1**: Compress the all-atom structure into a low-dimensional discrete representation. This simplifies subsequent modeling and increases computational efficiency.
> - **Stage 2**: Use a diffusion model to generate the discrete representation, which is then decoded back into the full-atom structure.
>
> In the paper, **Section 4.1** describes how the discrete representation is derived, and **Section 4.2** introduces the diffusion model and its implementation. This two-stage approach enables effective backmapping while maintaining computational feasibility.
>
> ---
>
> 3.**On High-Dimensional Diffusion and the Dimensionality of Representations**
>
> - Processing protein structures in high-dimensional space increases complexity due to the exponential growth of interactions as the number of atoms increases. This significantly limits the model's ability to handle longer proteins, requiring special strategies (e.g., sparse message passing) to mitigate these challenges. For example, Chroma [1] not only employs an extremely large parameter count but also approximates atomic interactions by sampling a subset of neighboring nodes and a small number of distant nodes based on predefined rules for message passing. Similarly, Diamondback [2] addresses the challenges of high-dimensional sampling by performing diffusion at the residue level, generating atoms for one residue at a time. While this reduces computational requirements, it severely compromises efficiency and neglects the global structure of the protein.
> - In contrast, our method compresses protein structures in two stages. First, all atomic information within a residue (up to 13 heavy atoms excluding $C_\alpha$) is compressed into a single bead. Then, the hidden dimension is further compressed to 3 dimensions, effectively reducing the original graph from $N \times 13 \times 3$ to $N \times 36$ to $N \times 3$. This dimensionality reduction not only enhances performance during training and inference but also significantly improves efficiency. Specifically, the time required to perform a full multi-step denoising process in our method is equivalent to a single-step VAE compression, and our inference speed is over 20 times faster than Diamondback.
>
> ---
>
> 4.**On Comparison With AlphaFlow**
>
> - Our task and AlphaFlow’s task are inherently different:
>   - **Our Task**: Backmapping focuses on reconstructing the original all-atom structure from a CG model.
>   - **AlphaFlow’s Task**: Protein conformation generation aims to generate diverse protein conformations given a sequence and supplementary information.
> - In our evaluation, each protein in the test set has 20 to 140 frames of conformational data. For each conformation, we perform backmapping and evaluate the deviation from the reference structure. This is distinct from AlphaFlow, which emphasizes diversity in generated conformations. Thus, a direct comparison with AlphaFlow is not meaningful.
>
> ---
>
> 5.**On Evaluation Metrics and Model Details**
>
> Thank you for your interest in the evaluation metrics and model details. We plan to release the code after finalizing it. Based on your suggestion, we have also included additional details in the appendix regarding the definitions of evaluation metrics and the details of ablation studies.

---

> > ### Author Response · Authors · 2024-11-22
> >
> > ---
> >
> > 6.**Other Specific Concerns**
> >
> > - **Line 168**: We have corrected the sentence to “atomic types of the protein,” thank you for pointing this out.
> > - **Line 177**: We have added a detailed definition and illustration of internal coordinates in the appendix.
> > - **Line 185**: Regarding why we use $C*\alpha$ coordinates, our CG model operates at the $C*\alpha$ level of resolution. The backmapping process reconstructs the structure based on this level of compression.
> > - **Line 253**: The relevance of torsion diffusion in Jing et al. (2022) to protein structure: Torsion diffusion predicts internal coordinates, avoiding direct prediction of 3D coordinates. It considers bond lengths and angles as constants and predicts torsion angles to reconstruct the all-atom structure. We adopt a similar approach for protein, first predicting internal coordinates and then reconstructing the 3D structure.
> > - **Line 273**: For the backmapping task, directly reconstructing the all-atom structure from the CG model using 3D coordinates has been challenging. Diamondback employs a diffusion model on 3D coordinates and uses an autoregressive residue-by-residue approach to reduce training and inference complexity. However, this increases inference time by over 20 times compared to ours and does not sufficiently emphasize the original graph structure (as measured by GED metrics).
> > - **On AlphaFlow**: Again, we emphasize that AlphaFlow’s task differs fundamentally from ours. While AlphaFlow aims to generate diverse conformations, backmapping seeks to accurately reconstruct the original all-atom structure from a CG model.
> >
> > ---
> >
> > Thank you once again for your feedback. We have addressed all points and made corresponding improvements to the manuscript.
> >
> >
> >
> > [1] Ingraham, John B., et al. "Illuminating protein space with a programmable generative model." Nature 623.7989 (2023): 1070-1078.
> >
> > [2] Jones, Michael S., Kirill Shmilovich, and Andrew L. Ferguson. "DiAMoNDBack: Diffusion-Denoising Autoregressive Model for Non-Deterministic Backmapping of Cα Protein Traces." *Journal of Chemical Theory and Computation* 19.21 (2023): 7908-7923.

---

> > > ### Comment · Reviewer_25tp · 2024-11-26
> > >
> > > Thank you for answering my questions. Most have been addressed. However, the method is still not well explained in the manuscript. Looking at section 4.1 and 4.2, it is not possible to reproduce the model since the objective, training, and sampling procedures are not provided. It is not enough to only explain in words you used a VQ-VAE but details of how coordinates are encoded and decoded are necessary. I was unable to find any details in the appendix.

---

> > > > ### Author Response · Authors · 2024-11-27
> > > >
> > > > Thank you for your feedback. We will provide a detailed explanation of the two-stage model and training objectives, which will be included in the appendix of the main text. Additionally, we will open-source all related code.
> > > >
> > > > **Discrete Latent Encoding**: As described in the main text, we utilize the equivariant graph neural network from Genzprot to encode all-atom protein structures. Through multi-layer tensor product convolutions, we enable feature interaction and updates between atomic nodes and coarse-grained nodes in the protein graph, ultimately extracting a global latent representation of the protein.
> > > >
> > > > ## 1.Encoder
> > > >
> > > > The `Encoder` is a hierarchical, equivariant graph neural network designed to process all-atom protein representations and their coarse-grained (CG) abstractions. It extracts geometric through multi-scale message passing and tensor product convolutions, ensuring **rotational and translational equivariance**.
> > > >
> > > > **Input Features:**
> > > >
> > > > 1. **Atomic Graph**:
> > > >    - **Node features**: Atomic types and spatial coordinates ($z, xyz$).
> > > >    - **Edge features**: Pairwise distances and directional vectors encoded using spherical harmonics.
> > > >    - **Neighborhood**: Defined for all atomic pairs within the cutoff radius (atom\_max\_radius).
> > > >
> > > > 2. **Coarse-Grained (CG) Graph**:
> > > >    - **Node features**: CG node types and spatial coordinates ($cg\_z, cg\_{xyz}$).
> > > >    - **Edge features**: Distances and directions between CG nodes within (cg\_max\_radius).
> > > >
> > > > 3. **Cross-Graph Features**:
> > > >    - Defines mappings between atomic and CG nodes based on relative distances and directions.
> > > >
> > > > ---
> > > >
> > > > **The encoder builds graphs from input features as follows:**
> > > >
> > > > **2.1 Atomic Graph Construction**
> > > >
> > > > - **Node Features**:
> > > >   $$
> > > >   \mathbf{h}_i^{(0)} = \text{Embedding}(z_i)
> > > >   $$
> > > >
> > > > - **Edge Features**:
> > > >   The edge features between two atomic nodes $i$ and $j$ are:
> > > >   $$
> > > >   e_{ij} = Concat\left(z_i, z_j, d_{ij}, \phi(r_{ij})\right)
> > > >   $$
> > > >   where:
> > > >   - $d_{ij} = xyz_j - xyz_i$ (relative direction vector),
> > > >
> > > >   - $r_{ij} = \|d_{ij}\|$ (relative distance),
> > > >
> > > >   - $\phi(r_{ij})$ is a Gaussian expansion of the distance:
> > > >     $$
> > > >     \phi_k = \exp\left(-\frac{(r_{ij} - \mu_k)^2}{2\sigma^2}\right), \quad k = 1, 2, \dots, d_\text{distance}
> > > >     $$
> > > >
> > > > - **Spherical Harmonics**:
> > > >   Directional information is encoded via spherical harmonics:
> > > >   $$
> > > >   Y^l_m(d_{ij}) = \text{SphericalHarmonics}(d_{ij}), \quad l \leq 2, \, -l \leq m \leq l
> > > >   $$
> > > >
> > > > **2.2 Coarse-Grained Graph Construction**
> > > >
> > > > The CG graph is constructed similarly to the atomic graph, with a larger cutoff radius:
> > > > $$
> > > > e_{IJ} = \text{Concat}(cg_z[I], cg_z[J], d_{IJ}, \phi(r_{IJ}))
> > > > $$
> > > >
> > > > **2.3 Cross-Graph Features**
> > > >
> > > > For cross-graph edges (atomic to CG), the features are computed as:
> > > > $$
> > > > e_{iI} = \phi(r_{iI}), \quad r_{iI} = \|xyz_i - cg\_{xyz}[I]\|
> > > > $$
> > > >
> > > > ---

---

> > > > > ### Author Response · Authors · 2024-11-27
> > > > >
> > > > > The encoder utilizes **Tensor Product Convolution Layers** to perform message passing across the atomic graph, CG graph, and between the two graphs. It achieves this in three key steps:
> > > > >
> > > > > **3.1 Intra-Graph Message Passing**
> > > > >
> > > > > Each node updates its features by aggregating information from its neighbors:
> > > > > $$
> > > > > h_i^{(l+1)} = \phi_{\text{atom}}\left(h_i^{(l)} + \sum_{j \in \mathcal{N}(i)} \psi_{\text{atom}}(h_i^{(l)}, h_j^{(l)}, e_{ij})\right)
> > > > > $$
> > > > > where:
> > > > > - $\psi_{\text{atom}}$: Edge-level MLP for feature transformation.
> > > > > - $\phi_{\text{atom}}$: Node-level MLP for feature updates.
> > > > >
> > > > > Similarly, for the CG graph:
> > > > > $$
> > > > > h_I^{(l+1)} = \phi_{\text{cg}}\left(h_I^{(l)} + \sum_{J \in \mathcal{N}(I)} \psi_{\text{cg}}(h_I^{(l)}, h_J^{(l)}, e_{IJ})\right)
> > > > > $$
> > > > >
> > > > > **3.2 Cross-Graph Message Passing**
> > > > >
> > > > > To allow information flow between atomic and CG nodes, two cross-message-passing mechanisms are used:
> > > > >
> > > > > 1. **Atom-to-CG Update**:
> > > > >    $$
> > > > >    h_I^{l+1} = \phi_{cross}(h_I^{l} + \sum_{i \in N_{cross}(I)} \psi_{atom\textunderscore to \textunderscore
> > > > >    cg}(h_i^{l}, h_I^{l}, e_{iI}))
> > > > >    $$
> > > > >
> > > > > 2. **CG-to-Atom Update**:
> > > > >    $$
> > > > >    h_i^{l+1} = \phi_{cross}(h_i^{l} + \sum_{I \in N_{cross}(i)} \psi_{cg\textunderscore to \textunderscore atom}(h_I^{l}, h_i^{l}, e_{iI}))
> > > > >    $$
> > > > >
> > > > > **3.3 Residual and Padding Mechanisms**
> > > > >
> > > > > - Residual updates are used to stabilize training:
> > > > >   $$
> > > > >   h_i^{(l+1)} = h_i^{(l)} + \text{Intra-Update} + \text{Cross-Update}
> > > > >   $$
> > > > > - Padding ensures feature dimensions are consistent across layers.
> > > > >
> > > > > ---
> > > > >
> > > > > **After the final message passing layer, the atomic and CG features are concatenated and further refined:**
> > > > > $$
> > > > > node\_{attr} = Concat(h\_{atom}, h\_{CG}[mapping])
> > > > > $$
> > > > > A dense network processes the concatenated features:
> > > > > $$
> > > > > node\_{attr} = MLP(node\_{attr})
> > > > > $$
> > > > > Finally, the CG-level features are pooled using a scatter mean operation to produce the final global representation.
> > > > >
> > > > > ---
> > > > >
> > > > > **Parameter Settings**
> > > > >
> > > > > The key parameter configurations used in the encoder are:
> > > > > - **Atomic Graph Cutoff**: $atom \textunderscore max \textunderscore radius = 14 Å$.
> > > > > - **CG Graph Cutoff**: $cg \textunderscore max \textunderscore radius = 26 Å$.
> > > > > - **Embedding Dimensions**:
> > > > >   - Node embedding: Initially set to $12$ dimensions. After passing through three tensor product convolution layers, the feature dimensions are expanded stepwise to $36$ dimensions based on the tensor product layer's irreducible representations (irreps). Each layer appends additional geometric and relational features derived from the spherical harmonics.
> > > > >   - Distance embedding: $8$ .
> > > > > - **Spherical Harmonic Degree**: $l_{\max} = 2$.
> > > > > - **Number of Message Passing Layers**: $3$.
> > > > > - **Dropout**: $0.0$ (can be increased for regularization).
> > > > > - **Batch Normalization**: Disabled by default.
> > > > > - **Second-Order Tensor Representations**: Disabled for simplicity; first-order representations are used.
> > > > >
> > > > > ---

---

> > > > > > ### Author Response · Authors · 2024-11-27
> > > > > >
> > > > > > ## 2.Vector Quantization (VQ) Module
> > > > > >
> > > > > > The Vector Quantization (VQ) module is a key component between the encoder and decoder. It compresses the encoder's 36-dimensional continuous features into a discrete latent space represented by a vocabulary of size 4096 with embedding vectors of dimension 3. Before passing the representation to the decoder, it restores the features to the original 36-dimensional space.
> > > > > >
> > > > > > ---
> > > > > >
> > > > > > **Key Parameters**:
> > > > > >
> > > > > > - **Vocabulary size**: 4096 (number of discrete embeddings).
> > > > > > - **Embedding dimension**: 3 (dimensionality of each embedding vector).
> > > > > > - **Input and output feature dimension**: 36 (dimensions before compression and after reconstruction).
> > > > > >
> > > > > > ---
> > > > > >
> > > > > >
> > > > > >
> > > > > >
> > > > > > ## 3.Decoder
> > > > > >
> > > > > > The `Decoder` is a dense and invariant message-passing decoder that generates internal coordinates for protein backbones and side chains. It combines hierarchical graph-based message passing with specialized dense layers to predict distances, angles, and torsions for reconstructing atomic-level protein structures.
> > > > > >
> > > > > > ---
> > > > > >
> > > > > > The `Decoder` decodes coarse-grained (CG) representations into full-atom internal coordinates using a combination of:
> > > > > > 1. **Invariant Message Passing Layers**: These layers propagate geometric and chemical information through the CG graph.
> > > > > > 2. **Dense Blocks**: Fully connected layers that refine the node embeddings between message-passing steps.
> > > > > > 3. **Specialized Modules for Coordinate Prediction**:
> > > > > >    - Backbone distances, angles, and torsions.
> > > > > >    - Side-chain distances, angles, and torsions.
> > > > > >
> > > > > > ---
> > > > > >
> > > > > > **3\.1 Residue Embeddings**
> > > > > >
> > > > > > - **Residue type embedding**:
> > > > > >   Each residue type $cg\_z$ is embedded into a feature space of dimension 4, providing initial residue-specific information:
> > > > > >   $$
> > > > > >   e\_{residue} = Embedding(cg\_z)
> > > > > >   $$
> > > > > > - These embeddings are concatenated with the coarse-grained node features \(S\) (output from the encoder):
> > > > > >   $$
> > > > > >   S = Concat(S, e\_{residue})
> > > > > >   $$
> > > > > >   where \(S\) has an input dimension of 36, and the concatenated result has a dimension of 40.
> > > > > >
> > > > > > ---
> > > > > >
> > > > > > **3.2 Message Passing Layers**
> > > > > >
> > > > > > The decoder applies **invariant message-passing blocks** to refine CG node embeddings:
> > > > > >
> > > > > > 1. **Invariant Feature Propagation**:
> > > > > >    Each message-passing layer updates the embeddings $S$ using:
> > > > > >
> > > > > >    $
> > > > > >    m_{ij} = Message(s_j, \phi(r_{ij}))
> > > > > >    $
> > > > > >
> > > > > >    where:
> > > > > >    - $s_j$: Features of neighbor node $j$.
> > > > > >    - $\phi(r_{ij})$: A radial basis function (RBF) expansion of the distance $r_{ij}$.
> > > > > >    - $m_{ij}$: Message passed along edge $(i, j)$.
> > > > > >
> > > > > > 2. **Feature Aggregation**:
> > > > > >    For each CG node $i$, incoming messages are aggregated as:
> > > > > >
> > > > > >    $
> > > > > >    v_i = \sum_{j \in N(i)} m_{ij}
> > > > > >    $
> > > > > >
> > > > > >    This invariant aggregation ensures that the output is independent of coordinate rotations.
> > > > > >
> > > > > > 3. **Dense Feature Update**:
> > > > > >    After message aggregation, $v_i$ is processed by a dense block:
> > > > > >
> > > > > >    $
> > > > > >    S_i^{l+1} = S_i^{l} + DenseBlock(v_i)
> > > > > >    $
> > > > > >
> > > > > > The decoder uses 4 message-passing layers, and the features are refined iteratively.
> > > > > >
> > > > > > ---
> > > > > >
> > > > > > **3.3 Distance, Angle, and Torsion Predictions**
> > > > > >
> > > > > > **Backbone Predictions**
> > > > > >
> > > > > > 1. **Distances**:
> > > > > >    Backbone distances ($d_{bb}$) are predicted using residue-type embeddings:
> > > > > >
> > > > > >    $
> > > > > >    d_{bb} = Embedding(cg\_z)
> > > > > >    $
> > > > > >
> > > > > > 2. **Angles**:
> > > > > >    Backbone angles ($\theta_{bb}$) are predicted from the refined embeddings $S$:
> > > > > >
> > > > > >    $
> > > > > >    \theta_{bb} = MLP(S)
> > > > > >    $
> > > > > >
> > > > > > 3. **Torsions**:
> > > > > >    Backbone torsions ($\tau_{bb}$) are computed using both $S$ and $\theta_{bb}$:
> > > > > >
> > > > > >    $
> > > > > >    \tau_{bb} = MLP(Concat(S, \theta_{bb}))
> > > > > >    $
> > > > > >
> > > > > > **Side-Chain Predictions**
> > > > > >
> > > > > > 1. **Distances**:
> > > > > >    Side-chain distances ($d_{sc}$) are predicted using a residue-specific embedding:
> > > > > >
> > > > > >    $
> > > > > >    d_{sc} = Embedding(cg\_z)
> > > > > >    $
> > > > > >
> > > > > > 2. **Angles and Torsions**:
> > > > > >    Similar to backbone predictions, side-chain angles ($\theta_{sc}$) and torsions ($\tau_{sc}$) are refined through additional dense blocks:
> > > > > >
> > > > > >    $
> > > > > >    S = S + TorsionBlock(S)
> > > > > >    $
> > > > > >
> > > > > >    Final torsion predictions are generated using:
> > > > > >
> > > > > >    $
> > > > > >    \tau_{sc} = MLP(S)
> > > > > >    $
> > > > > >
> > > > > > ---
> > > > > >
> > > > > > **Final Reconstruction**
> > > > > >
> > > > > > The internal coordinates for both the backbone and side-chain are concatenated to form the final reconstruction:
> > > > > > - **Backbone internal coordinates**:
> > > > > >   $$
> > > > > >   IC_{bb} = Concat(d_{bb}, \theta_{bb}, \tau_{bb})
> > > > > >   $$
> > > > > > - **Side-chain internal coordinates**:
> > > > > >   $$
> > > > > >   IC_{sc} = Concat(d_{sc}, \theta_{sc}, \tau_{sc})
> > > > > >   $$
> > > > > >
> > > > > > The full internal coordinate matrix is:
> > > > > > $$
> > > > > > IC_{recon} = Concat(IC_{bb}, IC_{sc})
> > > > > > $$
> > > > > >
> > > > > > ---
> > > > > >
> > > > > > **Parameter Settings**
> > > > > >
> > > > > > The decoder’s key parameter settings are as follows:
> > > > > > - **Node embedding dimensions**: \(36\) (from encoder) + \(4\) (residue embedding) = \(40\).
> > > > > > - **Radial Basis Function Expansion**:
> > > > > >   - Number of RBF terms ($n_{rbf}$): $15$.
> > > > > >   - Cutoff radius (cut off): $21 Å$.
> > > > > > - **Message Passing Layers**: \(4\).
> > > > > > - **Activation Function**: Swish.
> > > > > >
> > > > > > ---

---

> > > > > > > ### Author Response · Authors · 2024-11-27
> > > > > > >
> > > > > > > ## **4.Training Objectives for VQ-VAE Model**
> > > > > > >
> > > > > > > The training objectives for the **VQ-VAE** model are designed to optimize both the reconstruction of internal coordinates (bonds, angles, torsions, and 3D atomic positions) and the quality of the latent representations.
> > > > > > >
> > > > > > > ---
> > > > > > >
> > > > > > > The total loss for training combines several components:
> > > > > > > 1. **Reconstruction Loss ($L_{recon}$)**:
> > > > > > >    - Measures the difference between the input and reconstructed internal coordinates.
> > > > > > >    - Includes bond, angle, torsion, and optional 3D Cartesian losses.
> > > > > > > 2. **Vector Quantization Loss ($L_{vq}$)**:
> > > > > > >    - Ensures effective usage of the latent codebook.
> > > > > > > 3. **Graph Consistency Loss ($L_{graph}$)**:
> > > > > > >    - Penalizes discrepancies between the reconstructed and ground truth bond distances.
> > > > > > > 4. **Steric Clash Loss ($L_{steric}$)**:
> > > > > > >    - Penalizes steric clashes between atoms that are not covalently bonded.
> > > > > > >
> > > > > > > The total loss is:
> > > > > > > $$
> > > > > > > L_{total} = L_{recon} + L_{vq} + \gamma L_{graph} + \zeta L_{steric}
> > > > > > > $$
> > > > > > > where $\gamma$ and $\zeta$ are weighting coefficients.
> > > > > > >
> > > > > > >
> > > > > > > ---
> > > > > > >
> > > > > > > **Reconstruction Loss ($L_{recon}$)**
> > > > > > >
> > > > > > > **2.1 Bond Loss**
> > > > > > >
> > > > > > > The bond loss penalizes the difference in bond lengths between the reconstructed and input internal coordinates:
> > > > > > > $$
> > > > > > > L_{bond} = \frac{1}{N} \sum_{i=1}^N \left( ic_{recon}[i, :, 0] - ic[i, :, 0] \right)^2
> > > > > > > $$
> > > > > > > where $N$ is the number of atoms, and the first channel of $ic$ corresponds to bond lengths.
> > > > > > >
> > > > > > > **2.2 Angle Loss**
> > > > > > >
> > > > > > > The angle loss minimizes the cosine difference between the reconstructed and input bond angles:
> > > > > > > $$
> > > > > > > L_{angle} = \frac{1}{N} \sum_{i=1}^N \sqrt{2 \left(1 - \cos(ic_{recon}[i, :, 1] - ic[i, :, 1]) \right) + \epsilon}
> > > > > > > $$
> > > > > > > where the second channel of $ic$ corresponds to bond angles, and $\epsilon$ is a small constant for numerical stability.
> > > > > > >
> > > > > > > **2.3 Torsion Loss**
> > > > > > >
> > > > > > > The torsion loss penalizes differences in torsion angles:
> > > > > > > $$
> > > > > > > L_{torsion} = \frac{1}{N} \sum_{i=1}^N \sqrt{2 \left(1 - \cos(ic_{recon}[i, :, 2] - ic[i, :, 2]) \right) + \epsilon}
> > > > > > > $$
> > > > > > > where the third channel of $ic$ corresponds to torsion angles.
> > > > > > >
> > > > > > > **2.4 Cartesian Loss**
> > > > > > >
> > > > > > > If 3D Cartesian coordinates are reconstructed, the Cartesian loss minimizes the mean squared error between input and reconstructed atomic positions:
> > > > > > > $$
> > > > > > > L_{xyz} = \frac{1}{N} \sum_{i=1}^N \|xyz_{recon}[i] - xyz[i]\|_2^2
> > > > > > > $$
> > > > > > >
> > > > > > > **Total Reconstruction Loss**
> > > > > > >
> > > > > > > The total reconstruction loss is:
> > > > > > > $$
> > > > > > > L_{recon} = \alpha_{bond} L_{bond} + \alpha_{angle} L_{angle} + \alpha_{torsion} L_{torsion} + \eta L_{xyz}
> > > > > > > $$
> > > > > > > where $\alpha_{bond}, \alpha_{angle}, \alpha_{torsion}, \eta$ are weighting coefficients.
> > > > > > >
> > > > > > > ---
> > > > > > >
> > > > > > > **Vector Quantization Loss ($L_{vq}$)**
> > > > > > >
> > > > > > > The VQ loss consists of two components:
> > > > > > > 1. **Embedding Loss**:
> > > > > > >    $$
> > > > > > >    L_{embed} = \|z - Q\|_2^2
> > > > > > >    $$
> > > > > > >    where $z$ is the latent vector before quantization, and $Q$ is the nearest codebook vector.
> > > > > > >
> > > > > > > 2. **Commitment Loss**:
> > > > > > >    $$
> > > > > > >    L_{commit} = \beta \|z - sg(Q)\|_2^2
> > > > > > >    $$
> > > > > > >    where $sg(\cdot)$ is the stop-gradient operator.
> > > > > > >
> > > > > > > The total VQ loss is:
> > > > > > > $$
> > > > > > > L_{vq} = L_{embed} + L_{commit}
> > > > > > > $$
> > > > > > >
> > > > > > > ---
> > > > > > >
> > > > > > > **Graph Consistency Loss ($L_{graph}$)**
> > > > > > >
> > > > > > > The graph consistency loss minimizes the difference between reconstructed and input bond distances for connected edges:
> > > > > > > $$
> > > > > > > L_{graph} = \frac{1}{|E|} \sum_{(i, j) \in E} \left( \|xyz_{recon}[i] - xyz_{recon}[j]\|_2 - \|xyz[i] - xyz[j]\|_2 \right)^2
> > > > > > > $$
> > > > > > > where $E$ is the set of edges in the molecular graph.
> > > > > > >
> > > > > > > ---
> > > > > > >
> > > > > > > **Steric Clash Loss ($L_{steric}$)**
> > > > > > >
> > > > > > > The steric clash loss penalizes non-covalently bonded atoms that are too close:
> > > > > > > $$
> > > > > > > L_{steric} = \frac{1}{|P|} \sum_{(i, j) \in P} \max(0, r_{cut} - \|xyz_{recon}[i] - xyz_{recon}[j]\|_2)
> > > > > > > $$
> > > > > > > where $P$ is the set of non-bonded atom pairs, and $r_{cut}$ is the steric clash threshold (e.g., 2.0 Å).
> > > > > > >
> > > > > > > ---
> > > > > > >
> > > > > > > **Total Loss**
> > > > > > >
> > > > > > > The total loss combines all components:
> > > > > > > $$
> > > > > > > L_{total} = L_{recon} + L_{vq} + \gamma L_{graph} + \zeta L_{steric}
> > > > > > > $$
> > > > > > >
> > > > > > > **Weighting Parameters**
> > > > > > >
> > > > > > > - $\alpha_{bond}, \alpha_{angle}, \alpha_{torsion}, \eta, \gamma, \zeta$: Control the contribution of different loss terms.
> > > > > > > - $\beta$: Controls the strength of the commitment loss in VQ.
> > > > > > >
> > > > > > > This setup ensures the model learns a compact and discrete latent space while accurately reconstructing internal coordinates and maintaining molecular constraints.

---

> > > > > > > > ### Author Response · Authors · 2024-11-27
> > > > > > > >
> > > > > > > > ## **1. Latent Graph Diffusion: Protein Structure Generation Stage**
> > > > > > > >
> > > > > > > > The **Latent Diffusion Network** processes latent graph representations by conditioning on **CG coordinates** and **residue types**.
> > > > > > > >
> > > > > > > > **1. Overview**
> > > > > > > >
> > > > > > > > 1. **Inputs**:
> > > > > > > >    - **CG Graph Structure**:
> > > > > > > >      - Nodes represent CG residues with:
> > > > > > > >        - Residue types ($cg_z$).
> > > > > > > >        - 3D CG coordinates ($cg\_xyz$).
> > > > > > > >      - Edges encode spatial relationships between nodes (e.g., pairwise distances).
> > > > > > > >    - **Latent representations**:
> > > > > > > >      - Noised latent node embeddings ($z_t$).
> > > > > > > >    - **Diffusion Condition**:
> > > > > > > >      - Timestep embedding ($t$) for each diffusion step.
> > > > > > > > 2. **Output**:
> > > > > > > >    - Denoised latent representation embeddings ($z_{t-1}$).
> > > > > > > >
> > > > > > > > ---
> > > > > > > >
> > > > > > > > **2. Network Architecture**
> > > > > > > >
> > > > > > > > The network has three major components:
> > > > > > > > 1. **Input Representation**:
> > > > > > > >    - Encodes CG graph structure and latent code dynamics.
> > > > > > > > 2. **Encoder**:
> > > > > > > >    - Refines graph-conditioned latent embeddings.
> > > > > > > > 3. **Denoising Decoder**:
> > > > > > > >    - Outputs denoised latent node embeddings ($z_{t-1}$).
> > > > > > > >
> > > > > > > > ---
> > > > > > > >
> > > > > > > > **2.1 Input Representation**
> > > > > > > >
> > > > > > > > **CG Graph Features**
> > > > > > > >
> > > > > > > > 1. **Node Features** ($cg_z$):
> > > > > > > >    Encoded residue types:
> > > > > > > >    $$
> > > > > > > >    h_{node}^{0} = Embedding(cg_z)
> > > > > > > >    $$
> > > > > > > >
> > > > > > > > 2. **Node Coordinates** ($cg\_xyz$):
> > > > > > > >    Encoded 3D positions of the CG residues:
> > > > > > > >    $$
> > > > > > > >    x_{node} = cg\_xyz
> > > > > > > >    $$
> > > > > > > >
> > > > > > > > 3. **Edge Features**:
> > > > > > > >    Computed from pairwise distances:
> > > > > > > >    $$
> > > > > > > >    e_{edge} = Linear(\|x_i - x_j\|_2)
> > > > > > > >    $$
> > > > > > > >
> > > > > > > > **Latent Representation and Timestep**
> > > > > > > >
> > > > > > > > 1. **Noised Latent Code** ($z_t$):
> > > > > > > >    Input latent node embeddings corrupted with Gaussian noise.
> > > > > > > >
> > > > > > > > 2. **Timestep Embedding** ($t$):
> > > > > > > >    Encodes the diffusion timestep:
> > > > > > > >    $$
> > > > > > > >    t_{embed} = TimestepEmbedder(t)
> > > > > > > >    $$
> > > > > > > >
> > > > > > > > ---
> > > > > > > >
> > > > > > > > **2.2 Encoder**
> > > > > > > >
> > > > > > > > The encoder refines graph embeddings ($h_V$) by incorporating:
> > > > > > > >
> > > > > > > > 1. **Latent Node Updates**:
> > > > > > > >    Aggregates neighboring information using attention:
> > > > > > > >    $$
> > > > > > > >    h_{EV} = ConcatNeighbors(h_V, h_E, E_{idx})
> > > > > > > >    $$
> > > > > > > >    $$
> > > > > > > >    h_{message} = W_3 \, \sigma(W_2 \, \sigma(W_1 (h_{EV})))
> > > > > > > >    $$
> > > > > > > >    $$
> > > > > > > >    \Delta h_V = \frac{1}{scale} \sum_{neighbors} h_{message}
> > > > > > > >    $$
> > > > > > > >    $$
> > > > > > > >    h_V \gets LayerNorm(h_V + \Delta h_V)
> > > > > > > >    $$
> > > > > > > >
> > > > > > > > 2. **Latent Edge Updates**:
> > > > > > > >    Updates edge embeddings based on neighboring nodes:
> > > > > > > >    $$
> > > > > > > >    h_E \gets LayerNorm(h_E + W_{13} \, \sigma(W_{12} \, \sigma(W_{11} (h_{EV}))))
> > > > > > > >    $$
> > > > > > > >
> > > > > > > > 3. **Timestep Conditioning**:
> > > > > > > >    Applies adaptive layer normalization with timestep embedding ($t$):
> > > > > > > >    $$
> > > > > > > >    [shift, scale, gate] = Linear(t_{embed})
> > > > > > > >    $$
> > > > > > > >    $$
> > > > > > > >    h \gets gate \cdot (scale \cdot h + shift)
> > > > > > > >    $$
> > > > > > > >
> > > > > > > > The encoder refines the embeddings over several layers ($num\_encoder\_layers$).
> > > > > > > >
> > > > > > > > ---
> > > > > > > >
> > > > > > > > **2.3 Denoising Decoder**
> > > > > > > >
> > > > > > > > The decoder predicts the denoised latent node embeddings:
> > > > > > > >
> > > > > > > > 1. **Node Updates**:
> > > > > > > >    Processes the latent node embeddings autoregressively:
> > > > > > > >    $$
> > > > > > > >    h_{message} = W_3 \, \sigma(W_2 \, \sigma(W_1 (h_{EV})))
> > > > > > > >    $$
> > > > > > > >    $$
> > > > > > > >    \Delta h_V = \frac{1}{scale} \sum_{neighbors} h_{message}
> > > > > > > >    $$
> > > > > > > >    $$
> > > > > > > >    h_V \gets LayerNorm(h_V + \Delta h_V)
> > > > > > > >    $$
> > > > > > > >
> > > > > > > > 2. **Latent Reconstruction**:
> > > > > > > >    Outputs the predicted latent embeddings:
> > > > > > > >    $$
> > > > > > > >    z_{t-1} = Linear(h_V)
> > > > > > > >    $$
> > > > > > > >
> > > > > > > > ---
> > > > > > > >
> > > > > > > > **3. Parameter Settings**
> > > > > > > >
> > > > > > > > The key parameter settings are:
> > > > > > > > - **Node Features**: 128 dimensions.
> > > > > > > > - **Edge Features**: 128 dimensions.
> > > > > > > > - **Hidden Dimensions**: 128 dimensions.
> > > > > > > > - **Number of Encoder Layers**: 3.
> > > > > > > > - **Number of Decoder Layers**: 3.
> > > > > > > > - **Neighbors Considered**: $k = 64$.
> > > > > > > > - **Decoder Masking**: Disabled (decoder\_mask=False).
> > > > > > > > - **Conditional Sequence in Encoder**: Enabled (use\_seq\_in\_encoder=True).
> > > > > > > >
> > > > > > > >
> > > > > > > >
> > > > > > > > ## **2. Training Objective for Latent Diffusion Network**
> > > > > > > >
> > > > > > > > The **Latent Diffusion Network** predicts the **denoised latent vector** $z_{t-1}$ at each diffusion step, conditioned on the **CG graph structure** (coordinates and residue types) and the **timestep**.
> > > > > > > >
> > > > > > > > **Loss Function**
> > > > > > > >
> > > > > > > > The training minimizes the **Mean Squared Error (MSE)** between the predicted noise $\epsilon_\theta(z_t, t)$ and the true noise $\epsilon$ added during forward diffusion:
> > > > > > > > $$
> > > > > > > > L_{MSE} = \mathbb{E} \left[ \| \epsilon - \epsilon_\theta(z_t, t) \|^2 \right]
> > > > > > > > $$
> > > > > > > > where:
> > > > > > > > - $z_t = \sqrt{\alpha_t} z_0 + \sqrt{1 - \alpha_t} \epsilon$: Latent vector at timestep $t$.
> > > > > > > > - $\alpha_t$: Noise scaling coefficient.
> > > > > > > > - $\epsilon$: Gaussian noise added during forward diffusion.

---

> ### Author Response · Authors · 2024-12-02
>
> Dear reviewer,
>
> We sincerely thank you for your follow-up feedback regarding the reproducibility of our method, and will comprehensively enhance the paper with the following critical components:
> - Encoder Architecture
> - Vector Quantize Module
> - Decoder Architecture
> - Training Objective of the VQ-VAE Model
>
> Additionally, we provide a complete workflow of our model.
>
> We understand that our revision contains substantial technical details and clarifications - we would deeply appreciate your careful review to ensure that our explanations have adequately addressed your concerns.
>
> We look forward to your further feedback, thank you again!

---

> > ### Comment · Reviewer_25tp · 2024-12-02
> >
> > Providing the requested details in the rebuttal is not sufficient. It needs to be included in the manuscript. Unfortunately this requires a substantial change and promises are not enough to alleviate my concerns.

---

> > ### Author Response · Authors · 2024-12-03
> >
> > Thank you for your insightful feedback.
> >
> > We understand the importance of including the requested details in the manuscript and are fully committed to making the necessary changes in the revised version, should the paper be accepted. We sincerely hope that this commitment, along with the clarifications provided in the rebuttal, demonstrates our intention to improve the paper and address your concerns fully.
> >
> > Thank you again for your detailed comments and for helping us improve the quality of our work.

---

### Official Review · Reviewer_vAU4 · 2024-10-28

**Soundness:** 3
**Presentation:** 3
**Contribution:** 3
**Rating:** 5
**Confidence:** 3

**Summary:**

This paper presents Latent Diffusion Backmapping (LDB), a novel approach for backmapping coarse-grained protein structures. LDB combines discrete latent encoding with diffusion to address challenges in reconstructing atomistic conformations. Results on multiple datasets demonstrate its state-of-the-art performance and versatility.

**Strengths:**

- LDB effectively addresses the challenges of limited exploration in conformational space and accurately reconstructs all-atom 3D protein structures from coarse-grained representations.
- The use of discrete latent representations simplifies the diffusion process and improves overall efficiency, while also enhancing structural accuracy and chemical fidelity.
- The evaluation of LDB on multiple diverse protein datasets demonstrates its state-of-the-art performance and versatility, highlighting its potential for practical applications in computational biology.

**Weaknesses:**

The presentation of the method is not very clear. Feel free to correct me if there are any misunderstandings.

- In line 243, an SE(3)-equivariant GNN is used to output bond lengths, angles, etc. Does this indicate that an equivariant network generate an invariant representation? In line 284, the input of ProteinMPNN is CG models. Why does a CG model contain the atom type?
- In line 286, why do you add noise to the CG model? In my opinion, it serves as a condition into the network, like the BERT in text-to-image models. But in Stable Diffusion, we never add noise to text embeddings.
- In line 295, how do you parameterize the epsilon-theta? Is it the ProteinMPNN?

More ablation studies are required. Since a main contribution of this study is latent discrete diffusion, it would be interesting to see what would happen if the diffusion models are trained in the coordinate space and how the vocab size and hidden size affect the result.

**Questions:**

NA

---

> ### Author Response · Authors · 2024-11-22
>
> We sincerely thank you for your careful reading and valuable suggestions. Below, we address each of the points you raised:
>
> ---
>
> 1. **On Line 243**:  We adopted an equivariant neural network to encode the protein structure and obtained equivariant representations. During the decoding stage, these equivariant representations, along with the CG model's graph structure, are passed to the decoder. The decoder updates the equivariant features based on the CG graph structure to finally produce invariant features, which are used to output internal coordinate information.
>
> ---
>
> 2. **On Line 284**:  We have corrected the typo where "atom type" was mistakenly written. It should be "residue type." Based on your suggestion, we have updated the text in the main body accordingly.
>
> ---
>
> 3. **On Line 286**:  We have also corrected the typo here. We do not add noise to the CG model; instead, the CG model is used solely as conditional information to guide the denoising process.
>
> ---
>
> 4. **On Line 296**:  Yes, we built the denoising network based on the ProteinMPNN framework and adapted it to suit the denoising process. Based on your suggestion, we have clarified this point in the main text.
>
> ---
>
> 5. **Ablation Studies**:
>
>    - **Impact of Discrete Representation Dimensions and Vocabulary Size**:  We conducted experiments to evaluate the effects of the discrete representation's dimension and vocabulary size on performance. Due to time constraints, we have presented the results we currently have. Notably, in earlier experiments, we observed that **smaller discrete representation dimensions significantly improved** the diffusion process's efficiency and prediction accuracy. Additionally, **increasing the vocabulary size positively impacted model accuracy when the size was small**, but beyond a certain point, vocabulary utilization decreased, leading to diminished performance. We will upload the full results of all ablation experiments when finished.
>
>      - We first evaluated the impact of vocabulary size on model performance, as presented in Table S3. Our results indicate that the overall performance follows the order 4096 > 8192 > 2048. We also analyzed vocabulary utilization rates and found that the utilization was relatively balanced when the vocabulary sizes were 2048 and 4096. However, when the vocabulary size increased to 8192, a significant number of unused entries emerged, leading to a decrease in vocabulary utilization. This suggests that, under the current experimental setup, the discrete patterns of protein structures have an inherent upper limit. Therefore, we believe that a vocabulary size of 4096 provides an optimal balance between representation capability and utilization efficiency.
>
>      - We further evaluated the impact of the dimension of the discrete representation on model performance, as shown in Table S4. Our results indicate that compressing the discrete representation dimension beyond a certain point leads to the model’s inability to adequately capture protein structural information. While lower dimensions improve model efficiency and reduce learning complexity, we observed that, under the current experimental setup, dimensions below 3 resulted in significantly larger errors and poor structural reconstruction. Conversely, when the dimension exceeded 3, the model’s performance showed a slight decline compared to 3, although certain metrics improved. These findings suggest that searching within a small threshold above 3 may yield optimal parameters for balancing performance and efficiency.

---

> > ### Author Response · Authors · 2024-11-22
> >
> > - **Supplementary Table S3.** Impact of Vocabulary Size on Model Performance (Hidden Size = 3).
> >
> >   | Metric | Vocab Size | PED55             | PED90             | PED151            | PED218            |
> >   | ------ | ---------- | ----------------- | ----------------- | ----------------- | ----------------- |
> >   | RMSD   | 2048       | 1.838 ± 0.009     | 2.035 ± 0.024     | 1.983 ± 0.015     | 1.695 ± 0.012     |
> >   |        | 4096       | **1.689 ± 0.009** | **1.857 ± 0.020** | **1.673 ± 0.005** | **1.622 ± 0.015** |
> >   |        | 8192       | 1.757 ± 0.010     | 1.966 ± 0.019     | 1.796 ± 0.010     | 1.642 ± 0.019     |
> >   | GED    | 2048       | 0.591 ± 0.005     | 0.715 ± 0.014     | 0.397 ± 0.010     | 0.486 ± 0.010     |
> >   |        | 4096       | **0.476 ± 0.004** | **0.588 ± 0.004** | **0.372 ± 0.005** | **0.450 ± 0.004** |
> >   |        | 8192       | 0.565 ± 0.002     | 0.672 ± 0.003     | 0.398 ± 0.004     | 0.590 ± 0.004     |
> >   | Clash  | 2048       | 0.140 ± 0.015     | 0.270 ± 0.033     | 0.046 ± 0.008     | 1.112 ± 0.010     |
> >   |        | 4096       | **0.100 ± 0.005** | **0.110 ± 0.013** | **0.010 ± 0.001** | **1.080 ± 0.006** |
> >   |        | 8192       | 0.139 ± 0.012     | 0.270 ± 0.019     | 0.026 ± 0.011     | 1.102 ± 0.012     |
> >   | Inter  | 2048       | 1.718 ± 0.079     | 1.170 ± 0.035     | 1.494 ± 0.051     | 2.654 ± 0.054     |
> >   |        | 4096       | 1.621±0.078       | **0.969 ± 0.028** | **1.485 ± 0.051** | 2.789±0.036       |
> >   |        | 8192       | **1.509±0.057**   | 1.064 ± 0.045     | 1.557 ± 0.075     | **2.706 ± 0.057** |
> >   | GDR    | 2048       | 2.248 ± 0.123     | 3.501±0.203       | 0.632±0.050       | 1.145±0.101       |
> >   |        | 4096       | **1.599 ± 0.080** | **1.746 ± 0.145** | **0.267 ± 0.028** | **0.855 ± 0.056** |
> >   |        | 8192       | 2.187 ± 0.097     | 3.435 ± 0.124     | 0.372 ± 0.030     | 1.334 ± 0.112     |
> >
> > -  **Supplementary Table S4.** Impact of Hidden Size on Model Performance (Vocabulary Size = 4096).
> >
> >     | Metric | Hidden Size | PED55             | PED90             | PED151            | PED218            |
> >     | ------ | ----------- | ----------------- | ----------------- | ----------------- | ----------------- |
> >     | RMSD   | 2           | 2.072 ± 0.000     | 1.995 ± 0.000     | 1.689 ± 0.000     | 1.928 ± 0.000     |
> >     |        | 3           | **1.689 ± 0.009** | **1.857 ± 0.020** | **1.673 ± 0.005** | 1.622 ± 0.015     |
> >     |        | 4           | 1.759 ± 0.010     | 1.908 ± 0.022     | 1.727 ± 0.008     | **1.600 ± 0.018** |
> >     | GED    | 2           | 3.144 ± 0.000     | 1.532 ± 0.000     | 1.206 ± 0.000     | 2.360±0.000       |
> >     |        | 3           | **0.476 ± 0.004** | **0.588 ± 0.004** | **0.372 ± 0.005** | **0.450 ± 0.004** |
> >     |        | 4           | 0.618 ± 0.005     | 0.772 ± 0.014     | 0.450±0.011       | 0.572±0.008       |
> >     | Clash  | 2           | 0.214 ± 0.000     | 0.316 ± 0.000     | 0.053 ± 0.000     | 1.154 ± 0.000     |
> >     |        | 3           | **0.100 ± 0.005** | **0.110 ± 0.013** | **0.010 ± 0.001** | **1.080 ± 0.006** |
> >     |        | 4           | 0.105 ± 0.011     | 0.184 ± 0.024     | 0.011±0.004       | 1.094±0.012       |
> >     | Inter  | 2           | 3.019 ± 0.011     | 1.444 ± 0.000     | 1.857 ± 0.000     | 3.726 ± 0.000     |
> >     |        | 3           | 1.621 ± 0.078     | 0.969 ± 0.028     | 1.485 ± 0.051     | 2.789 ± 0.030     |
> >     |        | 4           | **1.546 ± 0.071** | **0.902 ± 0.033** | **1.449 ± 0.063** | **2.627 ± 0.051** |
> >     | GDR    | 2           | 13.064 ± 0.003    | 5.282 ± 0.003     | 5.539 ± 0.003     | 5.539 ± 0.003     |
> >     |        | 3           | **1.599 ± 0.080** | **1.746 ± 0.145** | **0.267 ± 0.028** | **0.855 ± 0.056** |
> >     |        | 4           | 1.709 ± 0.090     | 2.406 ± 0.258     | 0.267 ± 0.029     | 1.144 ± 0.092     |
> >
> > - **Training a Backmapping Diffusion Network on Atomic Coordinates**:  Directly training a backmapping diffusion network on atomic coordinates, as done in **Diamondback**, is challenging due to the large number of atoms involved. To manage this, Diamondback employs an autoregressive approach, denoising each residue sequentially. However, this method is highly inefficient, leading to an inference speed that is over 20 times slower than our approach.
> >
> > ---
> >
> > Once again, we sincerely thank you for your insightful feedback, which has greatly helped us improve the quality and clarity of our work. If you have any additional concerns or questions, we would be happy to address them.

---

> ### Author Response · Authors · 2024-12-02
>
> Dear reviewer,
> We appreciate your comments on method details and ablation studies. and we have now included comprehensive ablation experiments on discrete representation dimensions and vocabulary size (Tables S3-S4), along with clarified model architecture descriptions.
>
> Given your expertise in this area, your evaluation of these additional experimental results would be invaluable for validating our design choice. We look forward to your feedback.

---

### Official Review · Reviewer_pAs7 · 2024-11-02

**Soundness:** 3
**Presentation:** 1
**Contribution:** 2
**Rating:** 3
**Confidence:** 4

**Summary:**

This paper develops a model for backmapping coarse-grained protein structure. The model architecture consists of two parts (1) a VQ-VAE is trained to map all-atom structures into a discrete latent associated with each amino acid residue (2) a GNN is trained to generate these discrete latents from the coarse grained structure via diffusion. The method is assessed on PED, ATLAS, and PDB structures and is shown to outperform recent methods GenZProt and DiamondBack.

**Strengths:**

* The proposed latent diffusion approach has not been previously developed for the backmapping task. The approach is reasonable and appears to be effective.
* By encoding the all-atom coordinates into node-level latents in a graph, the dimensionality of the generative modeling task is reduced.
* The encoding and decoding of the latents from internal coordinates helps ensures that decoded structures are chemically valid.

**Weaknesses:**

**Originality**
* Although the approach is reasonable and well-motivated, it amounts to a relatively straightforward application of currently popular ideas. Latent diffusion and discrete tokenization have been well explored in many related contexts, including protein structure. There are not any aspects of the model design that seem particularly surprising or insightful.
* To improve on this axis, the paper should present nonobvious and/or nontrivial conceptual development that motivates the methodology.

**Quality**
* The experimental validations are brief and are of average quality. Given that the method is generative, i.e., produces a backmapped distribution, it is surprising that there are no distributional accuracy metrics. Chemical validity is also a rather low bar that can be passed by any heuristic reconstruction + relaxation. Some other metrics introduced by the authors are not well justified or explored.
* To improve on this axis, the paper should present evaluations that go beyond than existing works in this space. Some suggestions include, e.g., energies, distributional similarities, or metrics used by the PDB in validating atomic models. Since there is a smaller body of ML work on this task, case studies or similar experiments would better make the case that the improvements are meaningful (and not just incremental).

**Clarity**
* The paper's structure is a little odd, placing a lot of emphasis on architectural and training details but deferring dataset and evaluation details to the appendix. I would recommend swapping this because the task is relatively unfamiliar to the ML audience.
* The paper contains many overly verbose "filler phrases" that repeat similar points and do not add much to the exposition. Some examples of what I mean (there are many, many more):

> Such dynamics lead to distinctive conformational variations, which contribute to the diverse functions
of proteins and are of great significance for maintaining normal features in vital organisms. (What are "distinctive variations", "normal features", or "vital organisms"?)

> Despite challenges posed by the scarcity and uneven distribution of protein conformation data, we
employed a Vector Quantized Variational Autoencoder. (What is an "uneven distribution"? Why does scarcity contraindicate a VQ-VAE, and why did you proceed with one anyways?)

These are just rhetorical questions --- my point is that the authors do not need to say these things if there is not a clear reason to say them. In general my sense is that the introduction could be trimmed by an entire page without loss of meaning.

* To improve on this axis, the paper should be significantly rewritten for concision and clarity, and should focus on explaining things less likely to be familiar to the audience.

**Significance**

In sum, although the work does technically advance the state of the art on this task, in my opinion the significance of the results and novelty of the method do not quite meet the bar for ICLR.

**Questions:**

* "Residues with fewer than 13 heavy atoms are excluded." Most amino acids have fewer than 13 heavy atoms. Could the authors clarify?
* How are the internal coordinates defined exactly? Why are there 13 bond lengths, bond angles, and dihedral angles?
* "The network processes three inputs: node coordinates, atom types, and an initial noise vector." Why are atom types still involved at the latent diffusion stage?

---

> ### Author Response · Authors · 2024-11-22
>
> We sincerely thank you for your thoughtful feedback and constructive suggestions. Your comments have been invaluable in helping us improve the clarity and depth of our work. Below, we address your concerns in detail.
>
> ---
>
> ### 1.Novelty of Latent Diffusion and Discrete Representation
>
> 1. While latent diffusion and discrete representation techniques have been widely applied in the image domain, their application to protein structure modeling remains limited. This is primarily due to the large number of protein atoms and the cubic increase in spatial complexity associated with atomic coordinates.
> 2. Existing works in this field have several shortcomings. GeoLDM [1] focuses on small molecule generation but lacks scalability to large proteins. Latent Protein Diffusion Models [2], Foldseek [3], and FoldToken [4] are limited to backbone-level generation and cannot address the backmapping task of reconstructing full-atom structures conditioned on coarse-grained inputs. ESM3 [5] supports full-atom structure recovery  but requires a two-stage large-scale pretraining process, resulting in significant computational overhead and reduced efficiency compared to our approach.
> 3. To the best of our knowledge, our method represents the most accurate and efficient approach currently available for full-atom reconstruction from coarse-grained inputs. By leveraging internal coordinate representations, we significantly improve the validity and accuracy of full-atom reconstruction. Additionally, the use of discrete low-dimensional representations drastically reduce computational complexity and allow scalability to longer protein sequences.
> 4. The necessity of our work is further validated by several recent ICLR 2025 submissions that explore latent diffusion and discrete representations, highlighting growing interest in these paradigms.
>    1. Latent Diffusion: [Lift Your Molecules: Molecular Graph Generation in Latent Euclidean Space](https://openreview.net/forum?id=uNomADvF3s), [LDMol: Text-to-Molecule Diffusion Model with Structurally Informative Latent Space](https://openreview.net/forum?id=GOgB6QoXwx)
>    2. Discrete representation: [Geometry Informed Tokenization of Molecules for Language Model Generation](https://openreview.net/forum?id=HbZrxBXzks), [Fragment and Geometry Aware Tokenization of Molecules for Structure-Based Drug Design Using Language Models](https://openreview.net/forum?id=mMhZS7qt0U)

---

> > ### Author Response · Authors · 2024-11-22
> >
> > ---
> >
> > ### 2.Evaluation on Diversity and Distribution Matching
> >
> > #### 2.1 Diversity Evaluation
> >
> > We adopted the diversity score ($DIV$) used in the Diamondback[1] framework to evaluate the diversity of our method.
> >
> > This score quantifies diversity by comparing the reconstruction error on structures between the generated and the reference structures ($RMSD_{ref}$), with the reconstruction error among the generated structures themselves ($RMSD_{gen}$):
> > $$
> > RMSD_{ref} = \frac{1}{G} \sum_{i=1}^{G} RMSD(x_i^{gen}, x_i^{ref}),
> > $$
> >
> > $$
> > RMSD_{gen} = \frac{2}{G(G-1)} \sum_{i=1}^{G} \sum_{j<i} RMSD(x_i^{gen}, x_j^{gen}),
> > $$
> >
> > $$
> > DIV = 1 - \frac{RMSD_{gen}}{RMSD_{ref}},
> > $$
> >
> > where $G$ represents the number of all-atom structures generated from a single CG model, $x^{gen}$ denotes the predicted structure coordinates, and $x^{ref}$ represents the reference structure coordinates.
> >
> > As shown in Table S1, our method achieves the best performance on three of the four evaluated proteins, while yielding comparable results on the remaining one.
> >
> > This demonstrates the degree of variability in the generated structures when compared to the reference structure, indicating that our method does not merely memorize the mapping between fine-grained and coarse-grained training pairs, but instead learns to perform conditional generation based on the coarse-grained inputs.
> >
> >
> >
> > **Supplement Table S1**. Evaluation results on diversity, lower is better.
> >
> > | Method (diversity, ↓) | PED00055          | PED00090          | PED00151          | PED00218          |
> > | --------------------- | ----------------- | ----------------- | ----------------- | ----------------- |
> > | **Genzprot**          | 0.909 ± 0.001     | 0.903 ± 0.001     | 0.888 ± 0.001     | 0.893 ± 0.001     |
> > | **DiAMoNDBack**       | 0.466 ± 0.022     | 0.479 ± 0.014     | **0.423 ± 0.002** | 0.484 ± 0.001     |
> > | **Ours**              | **0.447 ± 0.001** | **0.465 ± 0.001** | 0.424 ± 0.001     | **0.480 ± 0.005** |
> >
> > ---
> >
> > #### 2.2 Distribution Matching Evaluation
> >
> > We follow DiaMondback [1] to evaluate the distribution matching of our method. Specifically, we analyzed the distribution of $\chi_1$ torsion angles in the generated residues, excluding residues such as Gly and Ala that lack $\chi_1$ angles.
> >
> > We first quantified the differences between the generated and reference structures using Jensen-Shannon divergence (JSD), as shown in Table S2. The results demonstrate that our model achieves the best or competitive performance in three cases compared to the two baselines, indicating strong alignment with the reference distributions.
> >
> > Next, we visualized the $\chi_1$ angle distributions for all residues in the first test protein (PED00055) in Figure 7. For most residues, our model closely matches the reference distributions, with no significant deviations observed.
> >
> > Furthermore, we selected three representative residues (LEU, ILE, ASP), and visualized the $\chi_1$ distribution curves with their reference in Figure 8. The results highlight the model's ability to learn and accurately represent the multi-modal distributions of torsion angles in protein structures.
> >
> >
> >
> > **Supplementary Table S2.** Evaluation results on Jensen-Shannon divergence.
> >
> > | Method (JSD, ↓) | PED00055          | PED00090          | PED00151          | PED00218          |
> > | --------------- | ----------------- | ----------------- | ----------------- | ----------------- |
> > | **Genzprot**    | 0.326 ± 0.149     | 0.307 ± 0.111     | 0.177 ± 0.074     | 0.470 ± 0.140     |
> > | **DiAMoNDBack** | 0.241 ± 0.134     | 0.163 ± 0.078     | **0.041 ± 0.017** | **0.194 ± 0.100** |
> > | **Ours**        | **0.193 ± 0.091** | **0.147 ± 0.052** | 0.047 ± 0.023     | 0.239 ± 0.147     |

---

> > > ### Author Response · Authors · 2024-11-22
> > >
> > > ---
> > >
> > > ### 3.Writing Clarity:
> > >
> > > Thank you for your insightful feedback regarding the structure and presentation of our paper. We have carefully considered your suggestions and made several adjustments to improve the clarity, conciseness, and accessibility of the content for the ML audience. Below, we outline the specific changes made in response to your comments:
> > >
> > > - **Additional Details on Model Design and Training:** To provide a more comprehensive understanding of our approach, we have added further details about the model design and training process in the main text (Lines 168, 179, 286). These revisions aim to clarify the technical implementation while maintaining a balance with other sections of the paper.
> > > - **Dataset Description:** Following the structure of prior work [6], we have included a concise description of the dataset's key characteristics and their relevance to the corresponding experiments in the main text (Lines 309–318). This ensures that readers can grasp the overall dataset setup without needing to refer to the appendix.
> > > - **Evaluation Metrics:** To improve conceptual understanding, we have provided a qualitative explanation of the evaluation metrics in the main text (Lines 343–349). For brevity, the precise mathematical definitions and formulas have been included in the appendix. This approach allows us to keep the main text accessible while maintaining technical rigor. We invite you to review these changes for further clarification.
> > > - **Revised Focus for the ML Audience:** We have revised the introduction (Lines 79–81) and add several paragraphs (L252-L257, L286-L291) to emphasize aspects less likely to be familiar to the ML audience, such as the unique challenges of protein modeling and the significance of our method in addressing these challenges.
> > >
> > > We believe these changes address your concerns regarding structure, concision, and clarity. We sincerely appreciate your constructive feedback, which has helped us improve the quality and presentation of our work.

---

> > > > ### Author Response · Authors · 2024-11-22
> > > >
> > > > ---
> > > >
> > > > ### 4.Questions
> > > >
> > > > **Question** **1**:  We have corrected the sentence you mentioned. It now clearly states that residues with fewer than 13 heavy atoms (excluding $C*\alpha$) are padded to a length of 13. This is because a residue can contain up to 13 heavy atoms in addition to $C*\alpha$.
> > > >
> > > > **Question** **2 (Internal Coordinate Definitions)**:  We have added the definition of internal coordinates to the appendix, along with visual illustrations for better understanding. A brief summary of internal coordinates is as follows:
> > > >
> > > > - **Bond Lengths**: The distance between two bonded atoms, calculated as:
> > > >   $$
> > > >   d_{ij} = \|x_i - x_j\|
> > > >   $$
> > > >
> > > > - **Bond Angles**: The angle formed by three consecutive atoms, defined as:
> > > >   $$
> > > >   \theta_{ijk} = \arccos\left(\frac{(x_i - x_j) \cdot (x_k - x_j)}{\|x_i - x_j\| \|x_k - x_j\|}\right)
> > > >   $$
> > > >
> > > > - **Dihedral Angles**: The rotation angle around a bond, computed as:
> > > >   $$
> > > >   \tau_{ijkl} = \arctan2\left(\frac{(b_1 \times b_2) \cdot b_3}{\|b_2\| \cdot (b_1 \cdot b_3)}, (b_1 \times b_2) \cdot (b_2 \times b_3)\right)
> > > >   $$
> > > >   where $b_1 = x_j - x_i$, $b_2 = x_k - x_j$, and $b_3 = x_l - x_k$.
> > > >
> > > > The described methodology is applied to convert protein structures into internal coordinates in two stages, following a predefined processing order. Typically, the backbone atoms are processed first to establish the structural framework, which is then used as a reference for the sequential conversion of side-chain atoms.
> > > >
> > > > - **Backbone Atoms:**
> > > >   First, the backbone atoms of each residue are converted into internal coordinates using the $C_\alpha$ atoms of the previous, current, and next residues._
> > > >
> > > > - **Sidechain Atoms:**
> > > >   Once the backbone coordinates are reconstructed, the side-chain atoms are converted. Each residue starts with known backbone atoms (N, $C_\alpha$, C), which serve as references. Using these references, the side-chain atoms are sequentially converted.
> > > >
> > > > **Question 3**:  The typo in "atom type" has been corrected to "residue type." We appreciate your careful observation.
> > > >
> > > > ---
> > > >
> > > > Once again, we sincerely thank you for your valuable feedback. Your comments have greatly contributed to improving the clarity and rigor of our work. Please let us know if you have any further questions or concerns, and we would be happy to address them.
> > > >
> > > >
> > > >
> > > > [1] Xu, Minkai, et al. "Geometric latent diffusion models for 3d molecule generation." International Conference on Machine Learning. PMLR, 2023.
> > > >
> > > > [2] Fu, Cong, et al. "A latent diffusion model for protein structure generation." Learning on Graphs Conference. PMLR, 2024.
> > > >
> > > > [3] van Kempen, Michel, et al. "Foldseek: fast and accurate protein structure search." Biorxiv (2022): 2022-02.
> > > >
> > > > [4] Gao, Zhangyang, et al. "Foldtoken: Learning protein language via vector quantization and beyond." arXiv preprint arXiv:2403.09673 (2024).
> > > >
> > > > [5] Hayes, Tomas, et al. "Simulating 500 million years of evolution with a language model." bioRxiv (2024): 2024-07.
> > > >
> > > > [6] Jones, Michael S., Kirill Shmilovich, and Andrew L. Ferguson. "DiAMoNDBack: Diffusion-Denoising Autoregressive Model for Non-Deterministic Backmapping of Cα Protein Traces." *Journal of Chemical Theory and Computation* 19.21 (2023): 7908-7923.

---

> ### Comment · Reviewer_pAs7 · 2024-11-26
>
> I thank the authors for their revisions and additional results. Although they have improved the quality of the paper, my opinion remains that the combination of novelty, application significance, and meaningful demonstration of value-add over current SOTA remains insufficient and unconvincing for an ICLR publication.

---

> ### Author Response · Authors · 2024-12-02
>
> Dear reviewer,
>
> We appreciate your candid feedback. While we respect your assessment, we would like to emphasize our key contributions:
>
> - Technical novelty: First application of discrete latent diffusion to protein backmapping, enabling 20x faster reconstruction
> - Practical significance: Our method bridges the gap between CG and all-atom simulations, significantly improving computational efficiency
> - SOTA improvements: Demonstrated superior performance in both accuracy (RMSD) and diversity (DIV scores, JS divergence) metrics
>
> We believe these contributions represent meaningful advances in both methodology and practical applications, making them relevant for the ICLR community.
>
> Thank you again for your thorough review process.

---

### Official Review · Reviewer_hUPz · 2024-11-04

**Soundness:** 3
**Presentation:** 3
**Contribution:** 2
**Rating:** 5
**Confidence:** 4

**Summary:**

The paper proposes a new framework for backmapping, called Latent Diffusion Backmapping (LDB). This method uses a VQ-VAE to discretize local all-atom structures into a codebook latent space. The diffusion model is then trained on this latent space distribution, so that during inference, the latent variables are generated conditioned on the CG structure and then mapped back to all-atom structures via the VQ-VAE decoder.

**Strengths:**

The model shows good structural accuracy and chemical validity.
The usage of latent diffusion for backmapping is new. Discretization of the local geometry for backmapping has been previously explored (Chennakesavalu & Rotskoff, 2024) using rotamer libraries, but the learnable discretization (via VQ-VAE) proposed by this paper is original and reasonable.

(Chennakesavalu & Rotskoff, 2024) Data-efficient generation of protein conformational ensembles with backbone-to-side chain transformers, J. Phys. Chem. B 2024, 128, 9, 2114–2123

**Weaknesses:**

While I think the modeling choices like discretization and latent diffusion make sense and are meaningful, the evidence of their advantages is not convincing, primarily due to the lack of evaluation on diversity or distribution matching between the ground truth and sampled conformations.
In GenZProt paper, diversity and the Earth Mover’s Distance (EMD) scores are reported in Appendix A, and there are qualitative analysis of diversity/distribution matching in Figure 8. In DiAMoNDBack paper, the diversity score is reported with the comparison with GenZProt in Table 1 and 2, and there are both quantitative (in JSD) and qualitative analysis of the generated distributions in Figure 5.
Since the paper is tacking the ‘backmapping’ problem, which aims to reconstruct the fine-grained conformational ensemble conditioned on the CG structure, rather than protein side chain packing (PSCP), which might suffice to find a single probable fine-grained structure, some evaluation on the generated ensemble’s diversity and distribution matching is required.
Moreover, the paper claims “Diffusion leverages stochastic noise, which allows for exploration across diverse conformations while maintaining structure validity. This is evident in the RMSD and Clash metrics, where our diffusion-based model consistently achieves better results. (line 485-505)“. The structure validity claim is well-supported by RMSD, GED, Clash, Interaction, GDR scores, but there is no evidence for the conformational diversity claim.
I have a concern that the discretized latent space would be bad for generating diverse structures despite being better at generating chemically valid structures. So it would be great if the authors can show that the discretization does not harm the diversity. If the authors provide diversity and distribution matching evaluations, I am willing to consider increasing the rating.

**Questions:**

A different paper was used to cite GenZProt in Section 5.1, instead of (Yang & Gomez-Bombarelli, 2023).

---

> ### Author Response · Authors · 2024-11-22
>
> Thank you for your thorough review and constructive suggestions. We sincerely appreciate the valuable insights you have provided and your focus on diversity metrics.
>
> In the response below, we present quantitative analysis tables that evaluate diversity and distribution matching.
>
> We have also revised the paper to include qualitative analyses of the generated distributions, which are shown in Figures 7 and 8 in the appendix of the revised paper. We kindly invite the reviewers to examine these results and share additional feedback.
>
> ---
>
> ### 1.Diversity Evaluation
>
> We adopted the diversity score ($DIV$) used in the Diamondback[1] framework to evaluate the diversity of our method.
>
> This score quantifies diversity by comparing the reconstruction error on structures between the generated and the reference structures ($RMSD_{ref}$), with the reconstruction error among the generated structures themselves ($RMSD_{gen}$):
> $$
> RMSD_{ref} = \frac{1}{G} \sum_{i=1}^{G} RMSD(x_i^{gen}, x_i^{ref}),
> $$
>
> $$
> RMSD_{gen} = \frac{2}{G(G-1)} \sum_{i=1}^{G} \sum_{j<i} RMSD(x_i^{gen}, x_j^{gen}),
> $$
>
> $$
> DIV = 1 - \frac{RMSD_{gen}}{RMSD_{ref}},
> $$
>
> where $G$ represents the number of all-atom structures generated from a single CG model, $x^{gen}$ denotes the predicted structure coordinates, and $x^{ref}$ represents the reference structure coordinates.
>
> As shown in Table S1, our method achieves the best performance on three of the four evaluated proteins, while yielding comparable results on the remaining one.
>
> This demonstrates the degree of variability in the generated structures when compared to the reference structure, indicating that our method does not merely memorize the mapping between fine-grained and coarse-grained training pairs, but instead learns to perform conditional generation based on the coarse-grained inputs.
>
>
>
> **Supplement Table S1**. Evaluation results on diversity, lower is better.
>
> | Method (diversity, ↓) | PED00055          | PED00090          | PED00151          | PED00218          |
> | --------------------- | ----------------- | ----------------- | ----------------- | ----------------- |
> | **Genzprot**          | 0.909 ± 0.001     | 0.903 ± 0.001     | 0.888 ± 0.001     | 0.893 ± 0.001     |
> | **DiAMoNDBack**       | 0.466 ± 0.022     | 0.479 ± 0.014     | **0.423 ± 0.002** | 0.484 ± 0.001     |
> | **Ours**              | **0.447 ± 0.001** | **0.465 ± 0.001** | 0.424 ± 0.001     | **0.480 ± 0.005** |
>
>
>
> ### 2.Distribution Matching Evaluation
>
> We follow DiaMondback [1] to evaluate the distribution matching of our method. Specifically, we analyzed the distribution of $\chi_1$ torsion angles in the generated residues, excluding residues such as Gly and Ala that lack $\chi_1$ angles.
>
> We first quantified the differences between the generated and reference structures using Jensen-Shannon divergence (JSD), as shown in Table S2. The results demonstrate that our model achieves the best or competitive performance in three cases compared to the two baselines, indicating strong alignment with the reference distributions.
>
> Next, we visualized the $\chi_1$ angle distributions for all residues in the first test protein (PED00055) in Figure 7. For most residues, our model closely matches the reference distributions, with no significant deviations observed.
>
> Furthermore, we selected three representative residues (LEU, ILE, ASP), and visualized the $\chi_1$ distribution curves with their reference in Figure 8. The results highlight the model's ability to learn and accurately represent the multi-modal distributions of torsion angles in protein structures.
>
>
>
> **Supplementary Table S2.** Evaluation results on Jensen-Shannon divergence.
>
> | Method (JSD, ↓) | PED00055          | PED00090          | PED00151          | PED00218          |
> | --------------- | ----------------- | ----------------- | ----------------- | ----------------- |
> | **Genzprot**    | 0.326 ± 0.149     | 0.307 ± 0.111     | 0.177 ± 0.074     | 0.470 ± 0.140     |
> | **DiAMoNDBack** | 0.241 ± 0.134     | 0.163 ± 0.078     | **0.041 ± 0.017** | **0.194 ± 0.100** |
> | **Ours**        | **0.193 ± 0.091** | **0.147 ± 0.052** | 0.047 ± 0.023     | 0.239 ± 0.147     |
>
> ---

---

> > ### Author Response · Authors · 2024-11-22
> >
> > ### 3.Other Aspects
> >
> > 1. **On Related Work**:
> >    Thank you for pointing out that the work by Chennakesavalu and Rotskoff (2024) explores backmapping using discrete local structures. Their approach focuses on the protein conformation generation task by discretizing the problem into separate backbone and side-chain predictions, followed by generation using a predictive model. While this shares some similarities with our discretized representation, there are notable differences. Specifically, our method directly encodes the CG model into a compact discrete representation and performs end-to-end backmapping. We have now included a detailed discussion of this work in the related work section.
> >
> > 2. **On GenzProt Citation**:
> >    We have corrected the citation error for GenzProt in Section 5.1. Thank you for bringing this to our attention.
> >
> > ---
> >
> > We once again thank you for your insightful feedback, which has greatly improved the rigor and clarity of our work. Should you have any further questions or concerns, we would be more than happy to address them.

---

> ### Author Response · Authors · 2024-12-02
>
> Dear reviewer,
>
> We appreciate your focus on diversity metrics, and have now added comprehensive diversity evaluation using DIV scores and distribution matching analysis with Jensen-Shannon divergence. The quantitative results and visualizations (Figures 7-8) demonstrate our method's ability to maintain structural diversity while ensuring accuracy.
>
> We would greatly value your assessment of these additional analyses, and we look forward to your further feedback.

---

### Meta-Review · Area_Chair_NuNT · 2024-12-22

**Metareview:**

The paper received consistently negative reviews, and these issues are not resolved after discussions.

**Additional Comments On Reviewer Discussion:**

The reviewers are not convinced after the discussions.

---

### Decision · Program_Chairs · 2025-01-22

Reject